# Deconstruction of the Ras switching cycle through saturation mutagenesis

Pradeep Bandaru[1,2,3], Neel H Shah[1,2,3], Moitrayee Bhattacharyya[1,2,3], John P Barton[4,5,6,7], Yasushi Kondo[1,2,3], Joshua C Cofsky[1,2,3], Christine L Gee[1,2,3], Arup K Chakraborty[4,5,6,7,8,9], Tanja Kortemme[10], Rama Ranganathan[11,12,13]*, John Kuriyan[1,2,3,14]*

[1]Department of Molecular and Cell Biology, University of California, Berkeley, Berkeley, United States; [2]California Institute for Quantitative Biosciences, University of California, Berkeley, Berkeley, United States; [3]Howard Hughes Medical Institute, University of California, Berkeley, Berkeley, United States; [4]Ragon Institute of MGH, MIT and Harvard, Cambridge, United States; [5]Department of Chemical Engineering, Massachusetts Institute of Technology, Cambridge, United States; [6]Department of Physics, Massachusetts Institute of Technology, Cambridge, United States; [7]Institute for Medical Engineering and Science, Massachusetts Institute of Technology, Cambridge, United States; [8]Department of Chemistry, Massachusetts Institute of Technology, Cambridge, United States; [9]Department of Biological Engineering, Massachusetts Institute of Technology, Cambridge, United States; [10]Department of Bioengineering and Therapeutic Sciences, California Institute for Quantitative Biomedical Research, University of California, San Francisco, San Francisco, United States; [11]Department of Pharmacology, University of Texas Southwestern Medical Center, Dallas, United States; [12]Green Center for Systems Biology, University of Texas Southwestern Medical Center, Dallas, United States; [13]Department of Biophysics, University of Texas Southwestern Medical Center, Dallas, United States; [14]Molecular Biophysics and Integrated Bioimaging Division, Lawrence Berkeley National Laboratory, Berkeley, United States

*For correspondence: rama.
ranganathan@utsouthwestern.edu
(RR); kuriyan@berkeley.edu (JK)

Competing interest: See
page 26

Reviewing editor: Alfonso
Valencia, Barcelona
Supercomputing Center - BSC,
Spain

**Abstract** Ras proteins are highly conserved signaling molecules that exhibit regulated, nucleotide-dependent switching between active and inactive states. The high conservation of Ras requires mechanistic explanation, especially given the general mutational tolerance of proteins. Here, we use deep mutational scanning, biochemical analysis and molecular simulations to understand constraints on Ras sequence. Ras exhibits global sensitivity to mutation when regulated by a GTPase activating protein and a nucleotide exchange factor. Removing the regulators shifts the distribution of mutational effects to be largely neutral, and reveals hotspots of activating mutations in residues that restrain Ras dynamics and promote the inactive state. Evolutionary analysis, combined with structural and mutational data, argue that Ras has co-evolved with its regulators in the vertebrate lineage. Overall, our results show that sequence conservation in Ras depends strongly on the biochemical network in which it operates, providing a framework for understanding the origin of global selection pressures on proteins.

## Introduction

Protein structures are remarkably robust to changes in primary sequence. This concept dates back to the early 1960s, when Max Perutz and John Kendrew observed striking similarities between the three-dimensional structures of myoglobin and hemoglobin, despite limited sequence identity (*Kendrew et al., 1954*; *Perutz et al., 1960*; *Lesk and Chothia, 1980*). An early landmark study concerning the mutational tolerance of proteins showed that λ-repressor, a bacterial transcription factor, can retain its ability to bind DNA and activate transcription despite many different mutations within the hydrophobic core (*Lim and Sauer, 1989*). More recently, studies using deep sequencing have reinforced the concept that proteins are remarkably tolerant of mutation while retaining the ability to fold and function (*Bershtein et al., 2006*; *Roscoe et al., 2013*; *Fowler and Fields, 2014*; *Tripathi and Varadarajan, 2014*; *Podgornaia and Laub, 2015*). In particular, 'two-hybrid' selection systems, combined with deep sequencing, have been utilized to determine the effects of mutations on the binding affinities of interacting proteins (*Dove et al., 1997*; *Joung et al., 2000*). This strategy has been used to probe the sensitivity to mutation of a PDZ domain-peptide interaction (*McLaughlin et al., 2012*). Residues in the PDZ domain are highly tolerant of mutation, except for those that govern folding, stability, and energetic interactions with the peptide target.

The view that proteins are tolerant of mutation contrasts with the observation that many metazoan proteins, particularly signaling proteins, are extraordinarily conserved in their sequences. Consider, for example, the Ras proteins, which are prototypical members of a superfamily of membrane-associated small G-proteins that control an extraordinary range of cellular processes (*Bourne et al., 1991*; *Rojas et al., 2012*). Ras proteins are capable of producing a diversity of signaling outputs by binding to effector proteins, most importantly to the Raf family of kinases, which activate the MAP kinase pathway (*Wittinghofer and Nassar, 1996*; *Wellbrock et al., 2004*). Ras proteins cycle between an active GTP-bound state that transduces signals by binding to effector proteins, and an inactive GDP-bound state that cannot bind to effectors (*Figure 1A*). The sequences of the three principal isoforms of Ras (H-Ras, K-Ras and N-Ras) are almost invariant in vertebrates, and Ras proteins are highly conserved across all metazoans (*Figure 1—figure supplement 1* shows a phylogenetic tree based on alignment of H-Ras orthologs) (*Johnson et al., 2017*).

Given that protein structures are tolerant to mutation, why are the sequences of Ras proteins (referred to collectively as 'Ras') so highly conserved? In particular, to what extent does the high degree of sequence conservation arise from the necessity to maintain the GTPase cycle that switches signaling on and off? The inherent sensitivity of Ras to activation by mutation is highlighted by the fact that it is an important oncogene (*Prior et al., 2012*). The signaling activity of Ras can be increased by mutations that prolong the GTP-bound state by reducing the rate of GTP hydrolysis, or that increase the rate at which GDP bound to Ras is replaced by GTP. The switch between the GDP- and GTP-bound states is accompanied by correlated conformational changes distributed throughout the structure of Ras (*Gorfe et al., 2008*; *Grant et al., 2009*; *Lu et al., 2016*). Two segments of Ras, referred to as Switch I and Switch II, undergo substantial rearrangements when GTP is replaced by GDP, as illustrated schematically in *Figure 1* (for a comprehensive review, see *Vetter and Wittinghofer, 2001*). Effector proteins, such as Raf and phosphatidyl inositol 3' kinase, transduce signals from Ras•GTP by binding to Switch I, and the conformation of the switch regions is also important for binding to GAPs and GEFs. We expect that the switching, enzymatic, and binding functions of Ras will impose stringent constraints on the sequence, but a systematic study of the nature of these constraints is lacking.

In this paper, we present an analysis of the constraints on the primary sequence of the GTPase domain of Ras that arise as a consequence of its function as a molecular switch that is turned off and on by GAPs and GEFs. To do this, we adapted the saturation point-mutagenesis and bacterial two-hybrid selection strategy that had been used to study sequence variation in PDZ domains (*Raman et al., 2016*). In this system, the binding of Ras•GTP to the Ras-binding domain (RBD) of C-Raf is coupled to the transcription of an antibiotic resistance factor, which enables the fitness of mutant forms of Ras to be evaluated through their effect on bacterial growth in the presence of an antibiotic. Co-expression of a GAP and a GEF enables the effect of regulators to be evaluated. The two-hybrid system omits some of the constraints that Ras normally operates under in a mammalian cell, such as interaction with the membrane (*Abankwa et al., 2008*; *Mazhab-Jafari et al., 2015*). Even without these constraints, our experiments show that the necessity of maintaining a properly

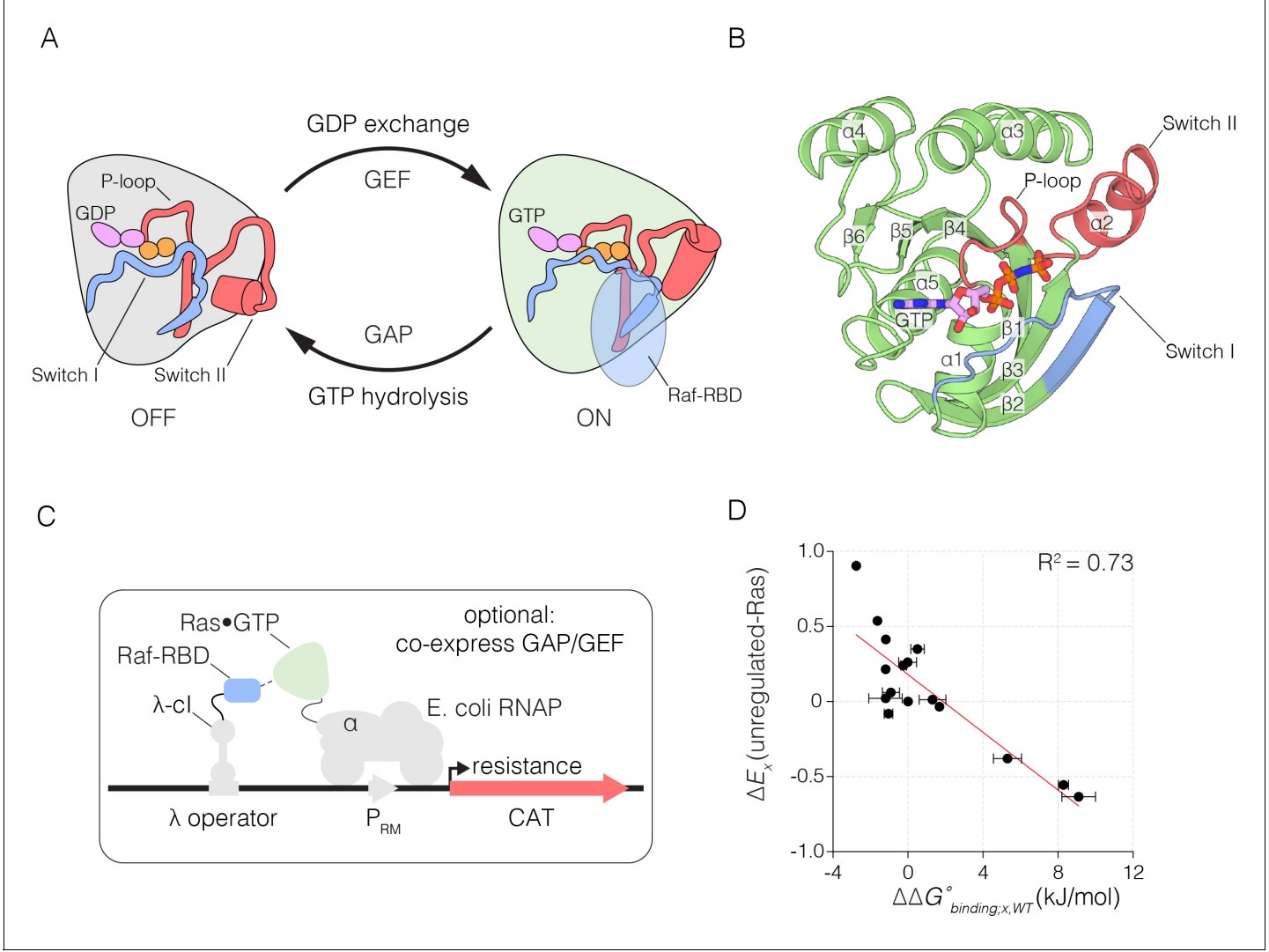

**Figure 1.** The Ras switching cycle and the bacterial two-hybrid system. (**A**) Ras cycles between an active, GTP-bound state and an inactive, GDP-bound state. Ras•GTP binds to effector proteins, such as Raf kinase, which binds to Switch I. The intrinsic hydrolysis of GTP is slow, unless catalyzed by a GTPase activating protein (GAP) which binds to Switch I. Intrinsic GDP release is also a slow process, unless facilitated by a guanine nucleotide exchange factor (GEF) which binds to both Switch I and Switch II. (**B**) Structure of Ras, highlighting secondary structure elements. (**C**) The bacterial two-hybrid system couples the Ras•GTP:Raf-RBD interaction to the production of an antibiotic resistance factor. The Ras variant library, the Raf-RBD, and the antibiotic resistance factor are encoded on three inducible plasmids. The GAP and GEF can also be co-expressed in the bacterial two-hybrid system. After protein expression, a fraction of the cells is removed and the plasmids encoding the Ras variant library are isolated and deep sequenced to count the frequency of each variant before antibiotic selection. The remainder of cells are subject to antibiotic selection with chloramphenicol and the plasmids encoding the Ras variant library are isolated and deep sequenced to count the frequency of each variant after antibiotic selection. The counts of each variant before and after selection are used to calculate the enrichment of each Ras variant. (**D**) In vitro validation of the bacterial two-hybrid system. The enrichment of individual Ras variants is approximately proportional to the change in Ras•GTP:Raf-RBD binding free energy upon mutation. Binding free energy of individual Ras mutants was measured by isothermal titration calorimetry, where error bars represent the standard deviation from three experiments, and relative enrichment values ($\Delta E_x$) are derived from wild-type Ras binding to Raf-RBD in the unregulated-Ras experiment.

The following figure supplements are available for figure 1:

**Figure supplement 1.** Conservation of the GTPase domains of H-Ras.

**Figure supplement 2.** Optimization of the bacterial two-hybrid system.

**Figure supplement 3.** Reproducibility of the bacterial two-hybrid system.

*Figure 1 continued*

**Figure supplement 4.** HPLC analysis of GMP-PNP loaded Ras.
**Figure supplement 5.** Raw ITC data for wild-type Ras.

regulated switching function severely constrains the sequence of Ras relative to that of proteins with a simpler binding function, such as λ-repressor or PDZ domains.

We find that for regulated wild-type H-Ras, in the presence of a GAP and a GEF, most mutations reduce the function of Ras to a mild extent, with the more severe effects being associated with mutations to residues that are critical for binding to GTP, the Raf-RBD and the regulators, and for the formation of the hydrophobic core. Strikingly, a distinct pattern of mutational sensitivity emerges for unregulated wild-type Ras, in the absence of the GAP and the GEF. Under these conditions, most mutations shift to being near-neutral and a set of gain-of-function mutations at hotspot residues appears, where almost any change to the wild-type residue results in increased activity. We show that these hotspots of activating mutations, which form a spatially contiguous network distributed around the GTPase domain, correspond to residues that reduce spontaneous switching through allosteric coupling to the GTP binding site. Mutations that activate Ras inappropriately would normally be eliminated by natural selection, and the combination of inactivating and activating mutations places constraints on nearly all of the residues in Ras. We also carried out a saturation mutagenesis experiment in the background of the oncogenic G12V mutation, and found that the presence of this mutation reduces the effect of mutations that activate wild-type H-Ras, indicative of coupling between the G12V mutation and the broadly distributed hotspot residues. Finally, we present an evolutionary analysis that, along with the mutational data, indicates that allosteric control in Ras was modified during the emergence of the vertebrate lineage. These data show that mutational sensitivity in Ras is strongly dependent on its regulatory network, a finding that can explain the high sequence conservation of Ras orthologs.

## Results and discussion

### A bacterial two-hybrid selection assay for Ras function

We engineered plasmids to inducibly express human H-Ras, the Ras-binding domain of human C-Raf (Raf-RBD), a GAP (the catalytic domain of human p120 RasGAP) and a GEF (the catalytic domain of human RasGRP1) in *E. coli*. The binding of Ras•GTP to Raf-RBD leads to transcription of an antibiotic resistance gene coding for chloramphenicol acetyltransferase (CAT). The catalytic domain of human H-Ras (residues 2 to 166, omitting the unique C-terminal segment that contains the membrane anchor) was fused to the α-subunit of *E. coli* RNA polymerase and Raf-RBD (residues 55 to 131) was fused to the phage λ-cI DNA binding domain (*Dove et al., 1997*; *Joung et al., 2000*). We refer to these fusion proteins as 'Ras' and 'Raf-RBD' in this paper, and these are expressed on separate inducible plasmids. Recruitment of these proteins to a third co-expressed plasmid encoding a λ-cI binding site stimulates the expression of CAT, leading to cell growth during selection with chloramphenicol. Expression of the catalytic domain of p120 RasGAP (*Scheffzek et al., 1997*), and the catalytic domain of the GEF RasGRP1 (*Iwig et al., 2013*), from the same plasmids that encode the Raf-RBD and CAT, respectively, drives the cycling of Ras by the GAP-stimulated hydrolysis of Ras•GTP to Ras•GDP, and the GEF-mediated exchange of GDP for GTP on Ras (*Figure 1C*).

We used the bacterial two-hybrid system to carry out four experiments, each of which involved analysis of a complete set of single-site mutations for the relevant variant of human H-Ras, expressed together with Raf-RBD. In the first experiment, variants of wild-type H-Ras were expressed in the presence of the GAP and the GEF (a condition we refer to as '*regulated-Ras*'). In the next experiment, H-Ras variants were expressed with the GAP, but without the GEF ('*attenuated-Ras*'). In the third experiment, variants of wild-type H-Ras were expressed without the GAP or the GEF ('*unregulated-Ras*'). Finally, we examined the effects of an oncogenic mutation by expressing variants of H-Ras that contain the G12V mutation, without the GAP or the GEF ('*Ras-G12V*').

For these experiments, we created two saturation point-mutagenesis libraries of Ras variants, one for wild-type H-Ras and one for the G12V variant, by site-directed mutagenic PCR, as described (*McLaughlin et al., 2012*), using primers containing randomized bases at each codon. The mutagenic primers contain NNS codons in the nucleotide sequence of Ras, where N represents a mixture of A/T/G/C and S represents a mixture of G/C. In this way, we created libraries in which the amino acid sequence of Ras is randomized one position at a time. The libraries were then cloned into the bacterial two-hybrid expression vector and deep-sequenced to ensure that each variant was represented in roughly equal proportion. *E. coli* cells were transformed with the libraries, screened in the bacterial two-hybrid system, and deep-sequenced before and after selection in order to calculate the effects of mutations in Ras.

## Quantifying the functional effects of Ras mutations

Chloramphenicol concentration and selection times were chosen to give maximal growth differences between oncogenic Ras mutants (e.g. G12V, Q61L) and mutants that are impaired in their ability to bind to Raf (e.g. T35S, D38N) (*Chuang et al., 1994*) (*Figure 1—figure supplement 2A*). Identical selection conditions were used for the four point-mutagenesis experiments. The ability of the two-hybrid system to report on Ras activity is illustrated in *Figure 1—figure supplement 2B*, which shows that the growth of bacteria expressing wild-type Ras decreases when an inactivating mutation is made to the GEF. In this case, Ras•GTP is hydrolyzed to Ras•GDP by the GAP, reducing CAT expression.

The bacterial two-hybrid system reports on the growth rate of cells containing an individual Ras variant, which we refer to as the 'fitness' of that variant. The effect on fitness of Ras mutations is quantified by the logarithm of the relative enrichment ($\Delta E_x$) of variants in the selected versus starting population, relative to wild-type (*WT*):

$$E_x = \ln\left[\frac{N_x(t)}{N_{0,x}(t)}\right]\ln\left[\frac{N_{WT}(t)}{N_{0,WT}(t)}\right]$$

Here, the subscripts $x$ and $WT$ refer to a particular Ras variant and to the wild-type or parent sequence, respectively. $N(t)$ is the number of cells at time $t$. We assume that the fitness of a Ras variant is directly related to the concentration of the Ras•GTP:Raf-RBD complex. $\Delta E_x$ values less than or greater than zero indicate a decrease or increase, respectively, in the steady-state concentration of Ras•GTP:Raf-RBD. All $\Delta E_x$ values reported here represent averages of two independent experiments, and the data were highly reproducible between duplicate experiments (*Figure 1—figure supplement 3*). Data for all experiments are available in *Supplementary file 1*.

We compared the relative fitness values, derived from two-hybrid experiments without the GAP or the GEF, to changes in the binding free energy for the interaction of Raf-RBD and Ras•GTP, measured using isothermal titration calorimetry (ITC). The ITC experiments were done for 25 Ras variants loaded with the non-hydrolyzable GTP analog, GMP-PNP. Complete loading of Ras with GMP-PNP was verified by HPLC analysis (*Figure 1—figure supplement 4*). The ITC data are consistent with earlier measurements of the affinity of the interaction between Raf-RBD and Ras variants (*Kiel et al., 2009*; *Rudolph et al., 2001*; *Wohlgemuth et al., 2005*) (*Figure 1—figure supplement 5*). Wild-type Ras binds to Raf-RBD with a dissociation constant ($K_D$) of ~170 nM, corresponding to a binding free energy ($\Delta G_{binding}$) of −38 kJ/mol. Many mutations in Switch I, such as D33N, T35S, and D38N, lead to a greater than 100-fold increase in the value of $K_D$ relative to wild-type, corresponding to values of $\Delta\Delta G_{binding}$ that range from 3 to 10 kJ/mol. There is a good correlation between the $\Delta E_x$ values and the corresponding values of $\Delta\Delta G_{binding}$ for mutations that have strong positive or negative effects on Raf-RBD binding (*Figure 1D*). For mutations with near wild-type binding, there is a greater variance of relative fitness, a result that might arise due to effects on GTP hydrolysis and exchange rates, which do not enter into the thermodynamic measurement.

For the saturation mutagenesis experiments, we found that the range of relative fitness values ($\Delta E_x$) in two of the experiments (attenuated-Ras and Ras-G12V) was different, when compared to the other two (unregulated-Ras and regulated-Ras). Although all four experiments were carried out using the same nominal conditions, the strength of the antibiotic selection appears to be somewhat variable from experiment to experiment. Differences in effective chloramphenicol concentration are expected to result in linear scaling of the relative enrichment values ($\Delta E_x$) in one dataset with respect

to the other. We applied scale factors of 1.2 and 0.7, respectively, to the $\Delta E_x$ values for the attenuated-Ras and Ras-G12V datasets. The scale factors were calculated such that the variance of $\Delta E_x$ values from the two datasets were equal for a reference residue (Val 112) in the hydrophobic core.

## Ras exhibits a global sensitivity to mutation when regulated by a GAP and a GEF

The values of the relative fitness, $\Delta E_x$, for every possible substitution in the sequence of Ras are shown in *Figure 2A* for the regulated-Ras experiment. The mode of the distribution is centered at a negative value of $\Delta E_x$, with most mutations resulting in a slight loss of function, with relatively few mutations resulting in a near-neutral effect or in a gain of function (*Figure 2B*). This is qualitatively different from the results of saturation point-mutagenesis of binding proteins such as a PDZ domain, and enzymes such as TEM-1 $\beta$-lactamase, where the distributions of fitness effects show a primary mode centered close to zero (*McLaughlin et al., 2012*; *Stiffler et al., 2015*). Our results suggest that regulation by the GAP and the GEF constrains Ras in a state where variations almost anywhere in the protein cause reduction of fitness. In principle, even a small loss of function can lead to effective purifying selection in evolving populations, and so the global constraint imposed by the regulators might explain the high degree of conservation in Ras sequences.

To identify the residues that are particularly sensitive to mutation, we averaged the fitness values for all the mutations for each residue. The distribution of residue-averaged $\Delta E_x$ values was fit to a normal distribution, and we define residues as highly sensitive to mutation if the corresponding residue-averaged $\Delta E_x$ value deviates more than 1σ from the mean value (*Figure 2C*). The highly sensitive sites are displayed on the structure of Ras in *Figure 2D*. These sites cluster within the hydrophobic core, and around the nucleotide binding site and the regions of Switch I and Switch II that interact with Raf-RBD, the GAP and the GEF. Most sites on the surface of Ras are less sensitive to mutation, except where surface residues are involved in interactions that are likely to stabilize the structure, as shown for an ion-pairing network involving sensitive residues that are surface exposed (*Figure 2E*).

As expected, mutations in Switch I that have been previously identified to impair Raf-RBD binding also compromise Ras function in our assay (*Spoerner et al., 2001*). The Ras•GTP:Raf-RBD binding interface includes interactions made by Glu 31, Asp 33, Glu 37, Asp 38, and Ser 39 in H-Ras. Nearly all mutations to these residues result in a loss of function, indicative of a highly constrained binding interface (*Figure 2—figure supplement 1A*). We note that since Raf-RBD and GAP have an overlapping interface, mutations in Switch I also impede GAP binding (*Scheffzek et al., 1997*; *Smith and Ikura, 2014*).

Many of the residues in Ras that coordinate the nucleotide, such as Lys 16, Thr 35, and Asp 119, are highly sensitive to mutation. Some of these mutations may cause protein instability, such as those at Asp 119. Interestingly, a few residues that interact with the nucleotide are not very sensitive to mutation, such as Lys 117 and Lys 147, and many mutations at these sites lead to a gain of Ras function. This may be due to the fact that the affinity of Ras for GTP is much higher than is required to saturate the active site (*Bourne et al., 1991*). Thus, residues that have only a modest effect on affinity do not alter the extent of GTP loading appreciably, given the high cellular concentrations of GTP (*Bennett et al., 2009*) (*Figure 2—figure supplement 1B*). Mutations that reduce GTP affinity may lead to fast-cycling Ras variants, which are able to more quickly replace GDP by GTP.

The binding and activity of the GEF (RasGRP1) imposes an additional constraint on the Ras sequence. One component of the interface between Ras and GEFs involves Switch II (*Boriack-Sjodin et al., 1998*; *Iwig et al., 2013*), where mutations to residues, including Asp 69, Tyr 71 and Arg 73, impair GEF binding. The binding footprint of the GEF extends beyond Switch II to residues 101–104, and mutations to residues in this region, such as Arg 102, cause a similar loss in Ras function. Not all mutations to residues in the GEF binding interface cause a loss of function. For example, Tyr 64 is crucial for GEF binding, but this residue is located in Switch II, near Gln 61, which is important for GTP hydrolysis. Thus, mutation of Tyr 64 and other residues in its vicinity leads to a complex pattern of mutational sensitivity in the regulated-Ras experiment (*Figure 2—figure supplement 1C*).

We tested the effect of a selected set mutations on GAP activity by using a fluorescent protein sensor for the detection of inorganic phosphate released by GTP hydrolysis, using purified mutant Ras proteins (*Brune et al., 1994*). Mutations in Switch I lowered GAP-catalyzed GTP hydrolysis rates, as expected (*Figure 2—figure supplement 2A*). We also tested the effect of Ras mutations on GEF

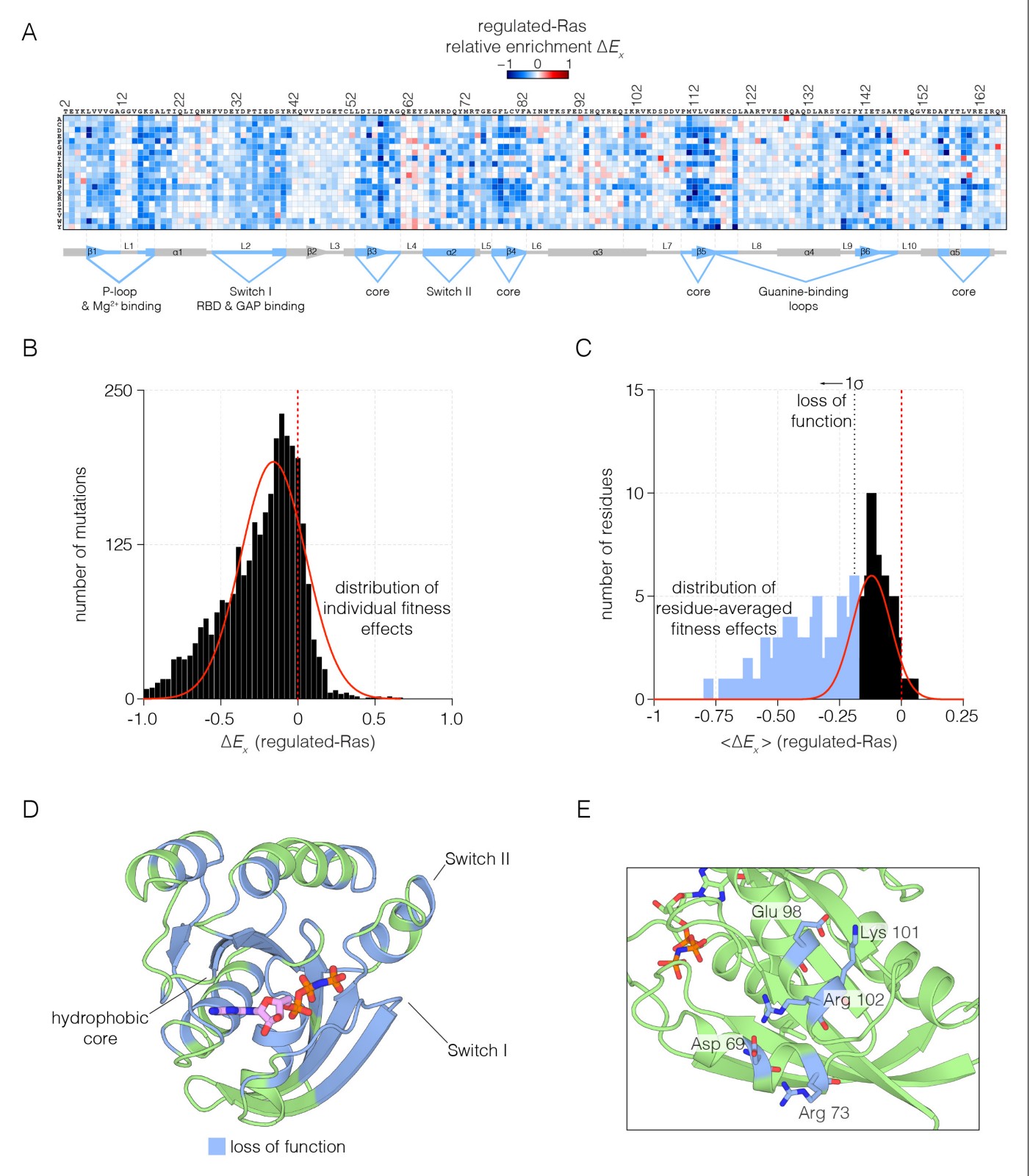

**Figure 2.** Mutational tolerance of Ras in the regulated-Ras experiment. (**A**) The results of the regulated-Ras experiment are shown in the form of a 165 × 20 matrix. Each row of the matrix represents one of the 20 amino acids, and each column shows one of the residues of Ras, from 2 to 166. Each entry in the matrix represents, in color-coded form, the value of the relative enrichment for the corresponding mutation ($\Delta E_x$). All data are normalized to the wild type Ras reference sequence, which has a relative enrichment value of zero. The numerical values of $\Delta E_x$ are provided in the supplementary data.
*Figure 2 continued on next page*

*Figure 2 continued*

(**B–C**) Distribution of individual fitness effects ($\Delta E_x$; left) and residue-averaged fitness effects ($<\Delta E_x>$; right). Residues with a significant loss of function effect on Ras ($<\Delta E_x>$ < 1σ from the mean) are indicated on the residue-averaged histogram. (**D**) Mapping the residues that lead to a significant loss of function onto the tertiary structure of Ras. These positions include the hydrophobic core, as well as residues involved in GDP/GTP and Raf-RBD binding. (**E**) Additional sites of mutational sensitivity include surface residues involved in ion-pairing networks that stabilize the GTPase fold.

The following source data and figure supplements are available for figure 2:

**Source data 1.** Raw data for GAP- and GEF- stimulated nucleotide hydrolysis and exchange rates.
**Figure supplement 1.** Mutational sensitivity of structural elements involved in nucleotide and effector binding.
**Figure supplement 2.** Validation of mutational effects through in vitro biochemical measurements of GAP- and GEF- stimulated nucleotide hydrolysis and exchange rates and a yeast growth assay.

activity, using an assay in which the release of labeled GTP or GDP from Ras is monitored as a function of time (*Eberth and Ahmadian, 2009*). Mutations in Switch II that altered fitness also affected GEF stimulated GDP exchange rates (*Figure 2—figure supplement 2B*). The numerical data are available in *Figure 2—source data 1*.

We used a yeast assay for the activity of human H-Ras to verify that a selected subset of the mutations that lead to decreased fitness in the two-hybrid assay do indeed lead to a loss of function. Transformation and expression of human H-Ras in the yeast *S. cerevisiae* renders the cells sensitive to heat shock. Mutations that affect the ability of Ras to bind nucleotide reduce Ras activity, restoring protection from heat shock, and consequently result in increased cell growth after heat shock (*Sass et al., 1986*). Using this assay, we verified that several mutations in the nucleotide binding regions that lead to decreased fitness also lead to increased yeast growth after heat shock. These results confirm that these mutations compromise the function of Ras (*Figure 2—figure supplement 2C*).

In summary, in the context of its natural regulatory network, Ras displays mutational sensitivity at nearly all positions. These positions include, but extend well beyond, regions known to mediate the basic biochemical activities of Ras, a result that highlights the value of deep mutational scanning in uncovering unexpected functional constraints in a mechanistically unbiased manner.

## The presence of the GAP without the GEF leads to strong Ras activation by oncogenic mutations

When Ras mutants are expressed in the presence of the GAP, but without the GEF (the attenuated-Ras experiment), we observe greater sensitivity to mutation across most of the structure (*Figure 3A*). It is likely that the presence of the GAP, without the compensating effect of the GEF, reduces the concentration of Ras•GTP to well below the dissociation constant for binding to Raf-RBD. In this situation, the effects of mutations that have a small deleterious effect are amplified, as a small extent of binding is further weakened by lower Ras•GTP concentration. In addition, the presence of the GAP without the GEF unmasks several strong gain-of-function mutations (*Figure 3B*). These mutations are much more strongly activating than in the regulated-Ras experiment, and nearly any substitution to residues involved in guanine or phosphate-binding and GTP hydrolysis lead to a gain of function, presumably because these interfere with the functioning of the GAP (*Figure 3C*).

In human cancers where Ras is mutated, 98% of the mutations occur at Gly 12, Gly 13 and Gln 61 (*Prior et al., 2012*; *Hobbs et al., 2016*). The first two residues are in the phosphate-binding P-loop of Ras, and Gln 61 plays a role in the catalysis of GTP hydrolysis, by positioning a water molecule for attack on the terminal phosphate group (*Pai et al., 1990*; *Scheffzek et al., 1997*). Under the conditions of the regulated-Ras experiment, mutations of Gly 12 and Gly 13 do not result in strong activation, and there is only a mild activating effect for mutations of Gln 61. In contrast, for the attenuated-Ras experiment, mutations at all three oncogenic sites are strongly activating (*Figure 3D*). Mutations of Gly 12 result in decreased GAP efficiency, and mutations at positions 13 and 61 affect GTP hydrolysis as well as the intrinsic rate of GDP-GTP exchange, as shown previously (*Mazhab-Jafari et al., 2015*).

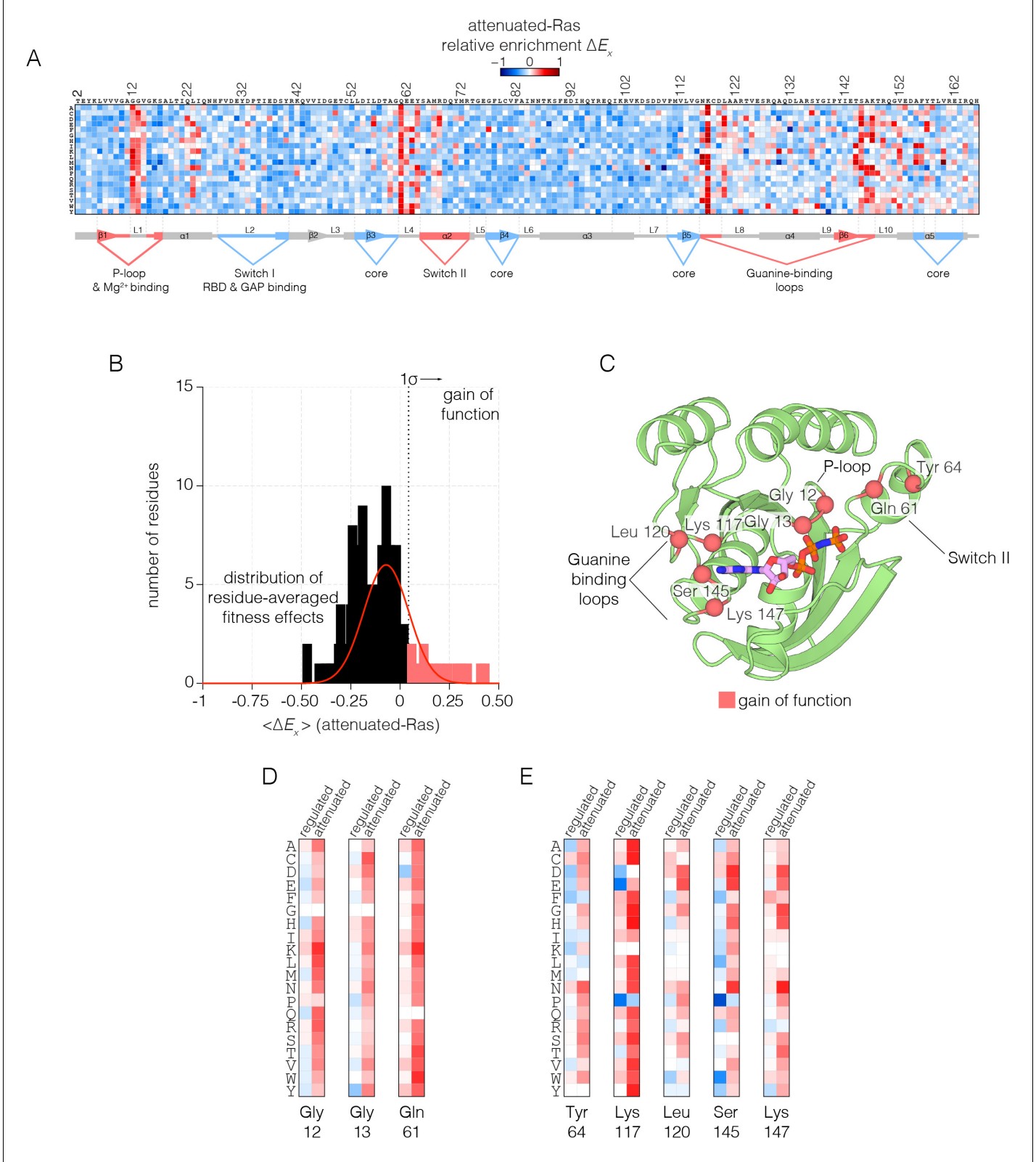

**Figure 3.** Mutational tolerance of Ras in the attenuated-Ras experiment. (**A**) Relative enrichment values ($\Delta E_x$) are shown in matrix form as in *Figure 2A*, for the attenuated-Ras experiment. Mutations at residues known to be mutated in human cancer (e.g. Gly 12, Gly 13, Gln 61, Lys 117) show a strong gain of function in this context. (**B**) Distribution of residue-averaged relative enrichment values. Residues with a significant gain of function effect on Ras (>1σ from the mean) are indicated on the histogram. (**C**) Mapping the residues that lead to a gain of function onto the tertiary structure of Ras. These

*Figure 3 continued on next page*

*Figure 3 continued*

positions span the P-loop, Switch II, and guanine binding loops of Ras and are directly involved in nucleotide coordination and hydrolysis. (D–E) Comparison of relative enrichment values for mutations at selected residues in the regulated-Ras and attenuated-Ras experiments. Substitutions at residues that are commonly mutated in cancer, such as Gly 12, Gly 13, and Gln 61, are exclusively gain-of-function in the attenuated-Ras experiment.

There are five other sites in Ras where almost any substitution leads to strong activation in the presence of the GAP without the GEF: Tyr 64, Lys 117, Leu 120, Ser 145 and Lys 147. Tyr 64, located near Gln 61, is located at the interface with the GAP (*Scheffzek et al., 1997*) (*Figure 3E*). The other four residues are involved in binding the guanine base of GTP, and mutation at these sites increases the intrinsic rate of nucleotide exchange (see below). The COSMIC database of somatic mutations in cancer includes mutations at some of these sites in Ras, such as Lys 117 and Leu 120, although these mutations are rare compared to the three principal sites of oncogenic mutation (*Bamford et al., 2004*).

## Hotspots of activating mutations in unregulated wild-type Ras

In our bacterial two-hybrid system, we have chosen experimental conditions such that wild-type Ras, without the GAP and the GEF, exhibits measurable growth in the presence of antibiotic selection (the unregulated-Ras experiment). Under these conditions, the balance of activation is set by the intrinsic rates of GTP hydrolysis and nucleotide exchange, and the expression level of Ras. The distribution of fitness effects for unregulated-Ras is remarkably distinct from that for regulated-Ras. Instead of global purifying selection (*Figure 2*), the distribution now displays a primary mode centered around zero, meaning that most mutations are near-neutral (see *Figure 2B*). This result demonstrates that in the absence of its local regulatory network, Ras behaves somewhat like other proteins studied through deep mutational scanning – generally tolerant to mutation, with a subset of positions showing significant functional effects.

An unexpected result is that for unregulated-Ras, there are many mutations, distributed throughout the structure, that result in *increased* fitness with respect to wild-type Ras (*Figure 4A*). For the majority of these residues, almost *any* mutation that alters the wild-type residue increases the activity of Ras in the unregulated-Ras experiment. We refer to residues where multiple mutations lead to activation as hotspots, and a subset of these residues were identified in the attenuated-Ras experiment (see *Figure 3A*).

Many of the hotspots of activation in the unregulated-Ras experiment involve residues for which activating mutations have not been previously identified in human Ras, and which do not lead to activation in the presence of the regulators (*Figure 4B* shows a scatter plot comparing the fitness effects of all mutations in the regulated-Ras and unregulated-Ras experiments). These conditionally-activating mutations are located further away from the active site than the mutations that activate in both experiments (*Figure 4B*). One example is Tyr 157, a residue located in helix α5, about 20 Å from the nucleotide-binding site and the switch regions. As can be seen in *Figure 4A*, replacement of Tyr 157 by 17 of the 19 possible alternatives results in increased function in the unregulated-Ras experiment. Another example is Gln 99, located in helix α3, on the side of Ras opposite to the location of Tyr 157, and also about 20 Å from the GTP-binding site. Replacement of Gln 99 by any residue, other than glycine and proline, leads to increased fitness in the unregulated-Ras experiment.

These hotspots of activation most likely correspond to residues that in the wild-type sequence play a critical role in maintaining the intrinsic rate of GTP hydrolysis, or suppressing the intrinsic rate of GDP release. We tested the effects on intrinsic GTP hydrolysis and nucleotide release for a subset of these mutations, using purified proteins and in vitro assays. Some of the mutations decrease the rate of hydrolysis, as seen for Q61L, E63P, and Q99A. Mutations at the base of Switch I (H27G), in helix α5 (Y157A), and in distal regions of the guanine binding loops (L120A), increase the rate of intrinsic nucleotide release (*Figure 4—figure supplement 1*).

Taken together, mutational scanning in regulated and unregulated Ras reveal a profound context dependence of the distribution of fitness effects. In the context of its regulatory network, Ras is subject to global purifying selection. However, without regulation, Ras displays a distinct character – general tolerance to mutation and the capacity for activating mutations that act allosterically to destabilize the inactive state.

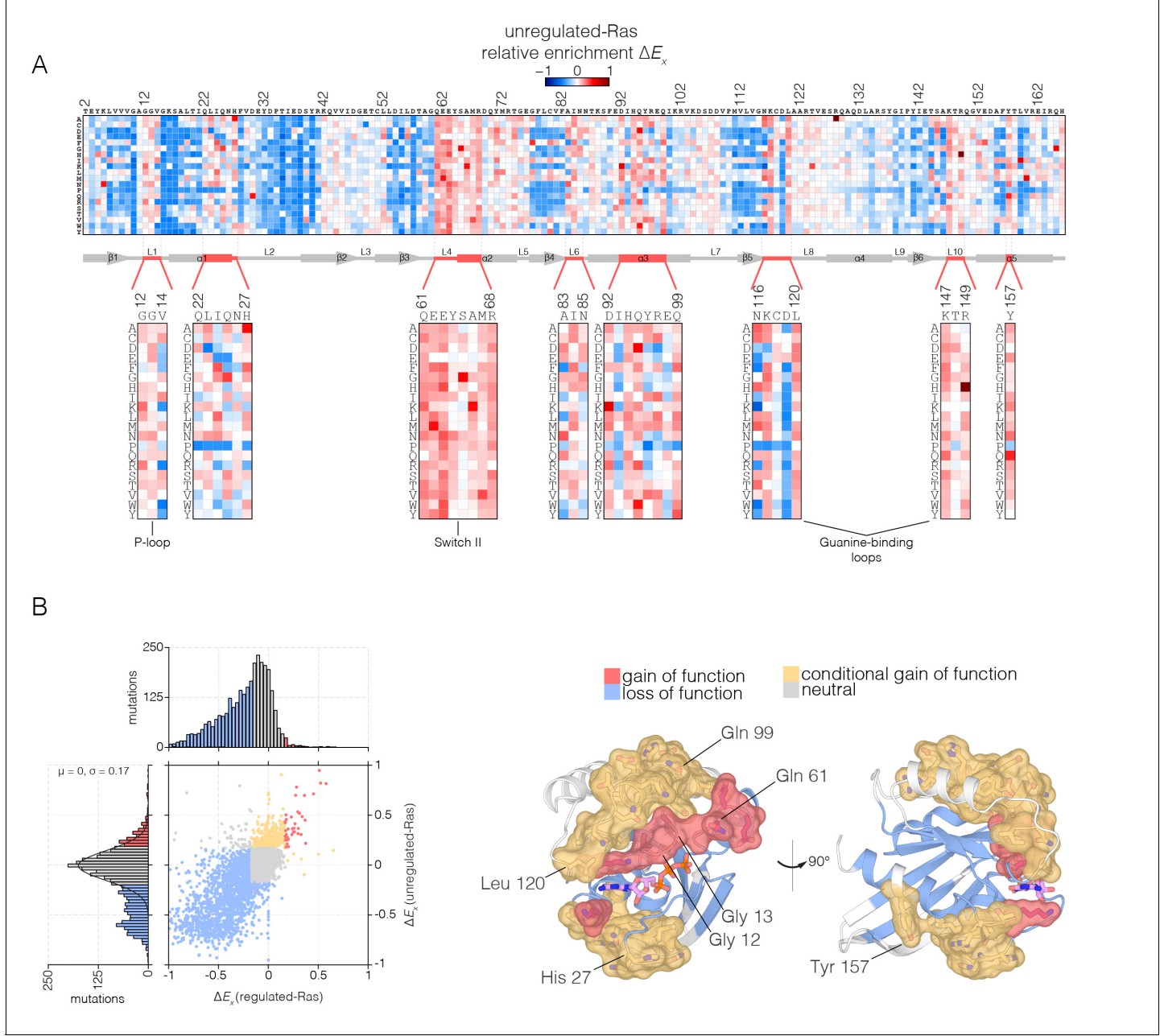

**Figure 4.** Mutational tolerance of Ras in the unregulated-Ras experiment. (A) Relative enrichment values ($\Delta E_x$) are shown in matrix form as in *Figure 2A*, for the unregulated-Ras experiment. In this experiment, Ras is expressed without the GAP and the GEF, and hotspots of activating mutations are revealed. Mutations at residues known to be mutated in human cancer (e.g. Gly 12, Gly 13, Gln 61, Lys 117) show a strong gain of function in this context. (B) (Left) A scatter plot of relative enrichment values ($\Delta E_x$) for the regulated-Ras and unregulated-Ras experiments. The distribution of relative enrichment values for each experiment are also shown. Loss-of-function mutations are shown in blue, and neutral mutations are shown in grey. Gain-of-function mutations in both experiments are shown in red, and mutations that are only gain-of-function in the unregulated-Ras experiment, but are neutral in the regulated-Ras experiment, are shown in yellow (conditional gain of function). (Right) The spatial distribution of gain-of-function (red) and conditional gain-of-function (yellow) mutations on the three-dimensional structure of Ras. Residues that contain a majority of gain-of-function mutations from the scatter plot in (b) are colored red, and residues that contain a majority of conditional gain-of-function mutations are colored in yellow.

The following figure supplement is available for figure 4:

**Figure supplement 1.** Intrinsic nucleotide release rates.

## The oncogenic G12V mutation attenuates the effects of mutations that activate Ras in the wild-type background

We carried out a saturation mutagenesis screen in the G12V background, without a GAP or a GEF (the Ras-G12V experiment), the results of which are shown in *Figure 5A*. The introduction of the G12V mutation into Ras markedly reduces the strongly activating effects of the hotspot mutations seen in the unregulated-Ras experiment (*Figure 5B*). This effect is seen for both residues that increase nucleotide exchange when mutated (such as Lys 117, Leu 120 and Lys 147; see *Figure 5C*), and residues that decrease the hydrolysis rate when mutated (such as Gly 13, Gln 61, and Gln 99) (*Janakiraman et al., 2010*; *Baker et al., 2013*).

The oncogenic G12V mutation is only mildly activating in the unregulated Ras experiment (see *Figure 4A*). The G12V mutation has two opposing effects on the activity of unregulated Ras. One is a decrease in the intrinsic rate of GTP hydrolysis with respect to wild type, which is an activating

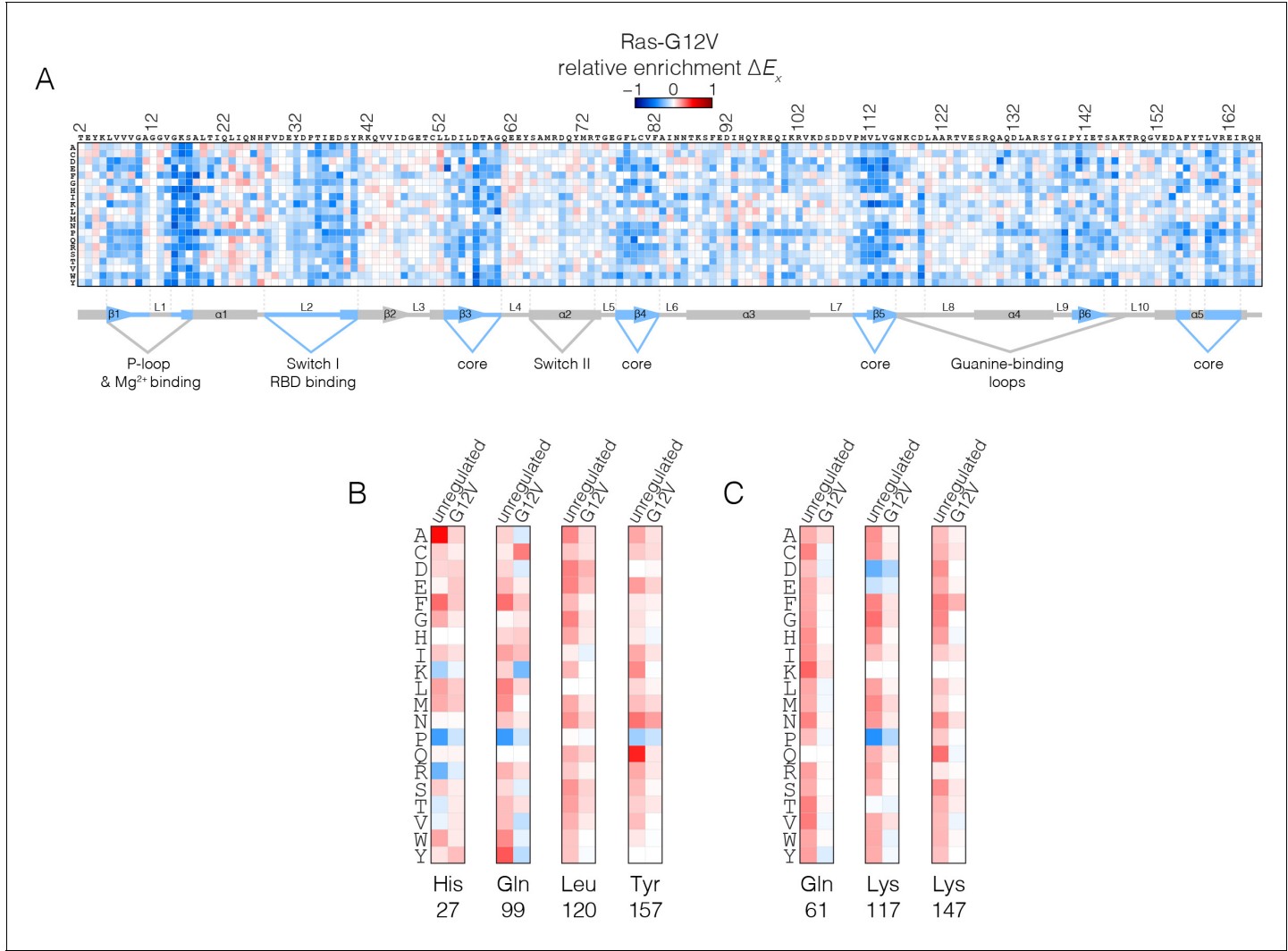

**Figure 5.** Mutational tolerance of Ras in the Ras-G12V experiment. (**A**) Relative enrichment values ($\Delta E_x$) are shown in matrix form as in *Figure 2A*, for the Ras-G12V experiment, in the absence of the GAP and the GEF. In this experiment, Ras shows a muted pattern of mutational sensitivity when compared to the wild-type unregulated-Ras experiment. Mutations at residues known to be mutated in human cancer (e.g. Gly 13, Gln 61, Lys 117) do not show a strong gain of function in this context. Mutations at Gly 12 are not included in the data. (**B–C**) Comparison of relative enrichment values for mutations at selected residues in the unregulated-Ras and Ras-G12V experiments. Substitutions at hotspot residues and residues that are commonly mutated in cancer, such as Gln 61, Lys 117, and Lys 147, result in a gain of function in the unregulated-Ras experiment, but have an attenuated effect in the Ras-G12V experiment.

effect in terms of Ras signaling. The phosphate-detection assay we use to measure GTP hydrolysis is not very sensitive when the rate of hydrolysis is low, and our data indicate only small reduction in hydrolysis rate for G12V versus wild-type. A much more sensitive NMR assay demonstrated, however, that the G12V mutation reduces the intrinsic hydrolysis rate substantially, to 10% of the wild-type rate (*Mazhab-Jafari et al., 2015*). The reduction in hydrolysis rate is counter-balanced by a reduction in the intrinsic rate of nucleotide exchange, by about 30% relative to wild-type in our assay (see *Figure 4—figure supplement 1*), with a similar reduction seen by NMR (*Mazhab-Jafari et al., 2015*).

Since Ras-G12V has a suppressed rate of GTP hydrolysis, we presume that mutations that further suppress this rate do not have a discernible effect on activity. Likewise, the decreased nucleotide exchange rate for Ras-G12V reduces the effect of mutations that tend to increase this rate. The epistasis between the effects of the oncogenic mutations and other mutations distributed across the structure of Ras will be an interesting line for future inquiry.

## Hotspot residues dampen the conformational dynamics of wild-type Ras

To address the structural mechanism of activating mutations in unregulated Ras, we ran molecular dynamics simulations of wild-type H-Ras and H-Ras with four hotspot mutations, introduced individually (H27A, Q99A, L120A, and Y157A). His 27 is located at the beginning of Switch I (see *Figure 4B*). Gln 99 is in helix α3, and an indication of its possible importance comes from the fact that it interacts with Switch II. Leu 120 is in the guanine-binding loop, with the sidechain in van der Waals contact with the edge of the nucleotide base. Tyr 157 is in helix α5, and the loop leading into the N-terminal portion of helix α5 packs against the guanine base. Mutations at each of these sites result in increased nucleotide release, and mutation at one of them (Q99A) also shows suppression of GTP hydrolysis (see *Figure 4—figure supplement 1*; note, however, that our assay for GTP hydrolysis does not provide a precise measurement of the suppression of the low intrinsic rate of GTP hydrolysis by Ras, as discussed earlier).

Molecular dynamics trajectories, each totaling 600 nanoseconds (ns), were generated for wild-type Ras and each mutant. In each case, we began with the crystal structures of wild-type H-Ras bound to GTP or GDP (PDB codes 5P21 and 4Q21, respectively [*Milburn et al., 1990*; *Pai et al., 1990*]), replacing the appropriate sidechains by alanine for each simulation of a mutant protein. Some of the trajectories were generated as one contiguous 600 ns simulation, while others were generated in two independent blocks of 300 ns each, and then merged for analysis.

In each of the simulations of the mutant proteins bound to GTP, we discern changes in the dynamics of the switch regions, the guanine-binding loops and helix α3, which abuts Switch II. A superposition of structures sampled from the GTP-bound simulations is shown in *Figure 6*. The overall structure of Ras is very stable in each of the simulations, as can be seen in views of the superimposed structures facing the side of Ras that is opposite to the switch regions (middle diagrams in each of the panels in *Figure 6* and *Figure 6—figure supplement 1*, with helices α4 and α5 facing the viewer). Excluding the switch regions, the r.m.s. fluctuation in Cα positions from the mean structure is very small (less than 1 Å) over the course of each of the trajectories, and the r.m.s. deviation from the crystal structure is also small (less than 1 Å). Comparison of the GDP-bound simulations was less informative, since the increased dynamics of the switch regions in these simulations made it difficult to identify the effects of the mutations.

Switch II is very mobile in the simulation of wild-type Ras•GTP, with r.m.s. fluctuations in Cα positions of ~1.2 Å when the structures are superimposed without using Switch I and II. But in all four simulations of the mutant structures, there are increased conformational fluctuations of Switch II (left and right diagrams in each of the panels in *Figure 6*; the r.m.s. fluctuations in Cαpositions range from ~1.4 Å to ~1.7 Å (*Figure 6—figure supplement 2*). This is particularly striking since the four mutations are at diametrically opposite ends of the structure, and the changes in dynamics indicate allosteric coupling between distant points in Ras and the Switch II helix. There are also changes in the dynamics of helix α3, which is structurally coupled to Switch II. Alterations in the guanine-binding loops and Switch I are also evident in the trajectories, but are more difficult to appreciate in the structural superpositions.

The molecular dynamics trajectories suggest that there is a loosening of the structure of Ras when His 27, Gln 99, Leu 120, and Tyr 157 are mutated. Similar effects were seen in a molecular

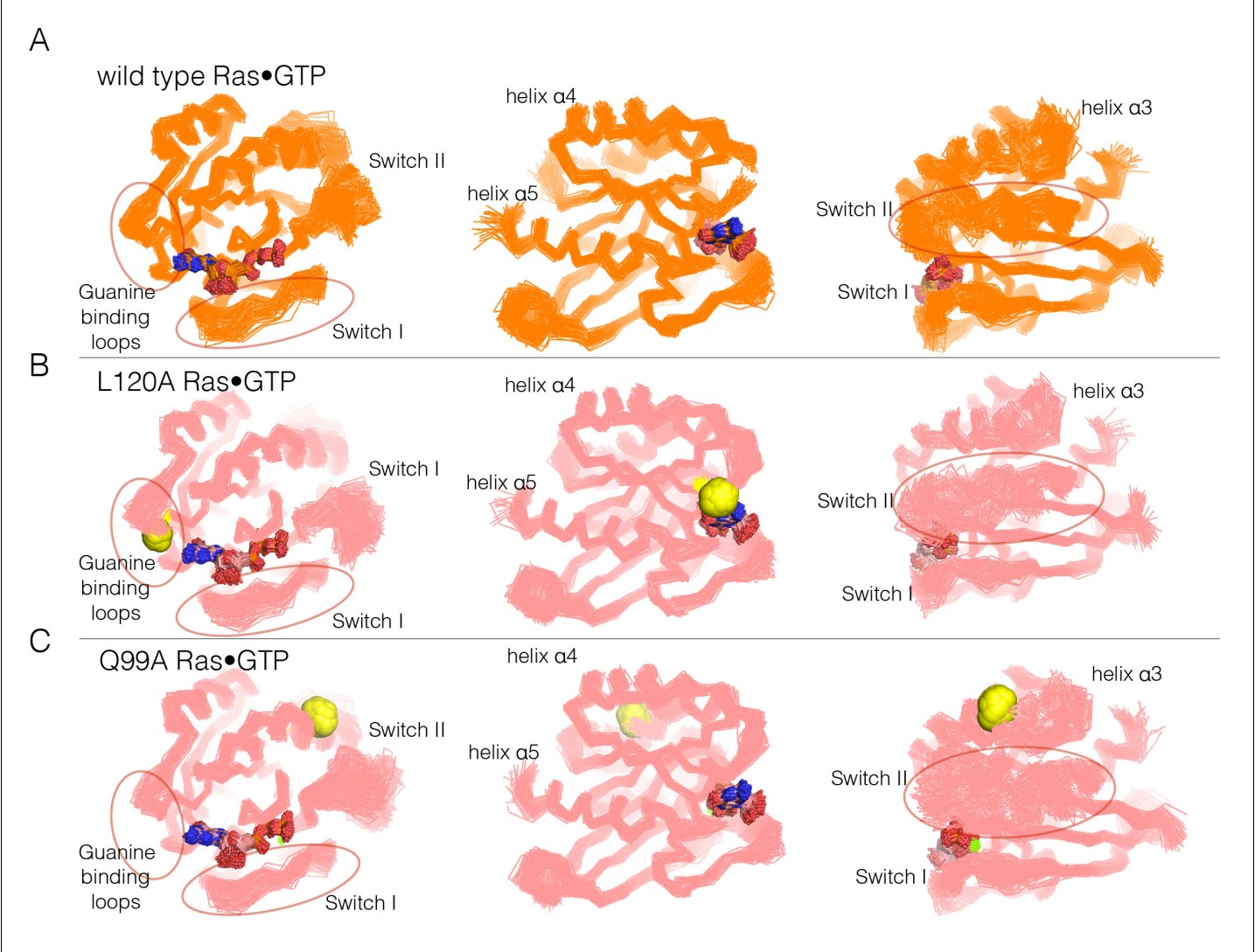

**Figure 6.** Superposition of structures from molecular dynamics simulations of GTP-bound forms of Ras. (**A**) Wild-type Ras•GTP. The diagrams show the superposition of 30 structures sampled evenly from two 300 ns trajectories. The diagram on the left shows a canonical view of Ras•GTP. The two other diagrams show two orthogonal views. (**B**) As in (**A**), for Ras•GTP L120A, shown as yellow spheres. (**C**) As in (**A**), for Ras•GTP Q99A, shown as yellow spheres.

The following figure supplements are available for figure 6:

**Figure supplement 1.** Superposition of structures from GTP-bound simulations of wild type, H27A, and Y157A mutants.

**Figure supplement 2.** Flexibility in Ras measured by root mean squared fluctuations of Cα atoms.

dynamics analysis of the effects of oncogenic mutations in K-Ras (*Lu et al., 2016*). To assess the effects of mutations quantitatively, we carried out an analysis of interaction networks in the simulations, as has been done previously for a Gln 61 mutant of Ras (*Fetics et al., 2015*). We used a different method from that used in the previous simulation study, and we determined the strength of a noncovalent interaction between sidechains by counting the number of atom pairs between residues within a 4.5 Å cutoff, normalized by the sizes of the interacting residues (*Brinda and Vishveshwara, 2005*; *Ghosh and Vishveshwara, 2007*; *Bhattacharyya et al., 2013*, *2016*). Interactions between sidechains that are present in more than a threshold fraction of instantaneous structures in the trajectory are identified as stable interactions. We chose 50% as the threshold value for definition of a

stable contact, and analyzed each trajectory by drawing lines between the Cα atoms of the corresponding residues in stable contacts, as shown for the simulation of wild-type Ras•GTP in *Figure 7A*. The stability of the wild-type Ras simulation is reflected in the large number of stable contacts that anchor all of the neighboring structural elements, throughout the structure (*Figure 7A*).

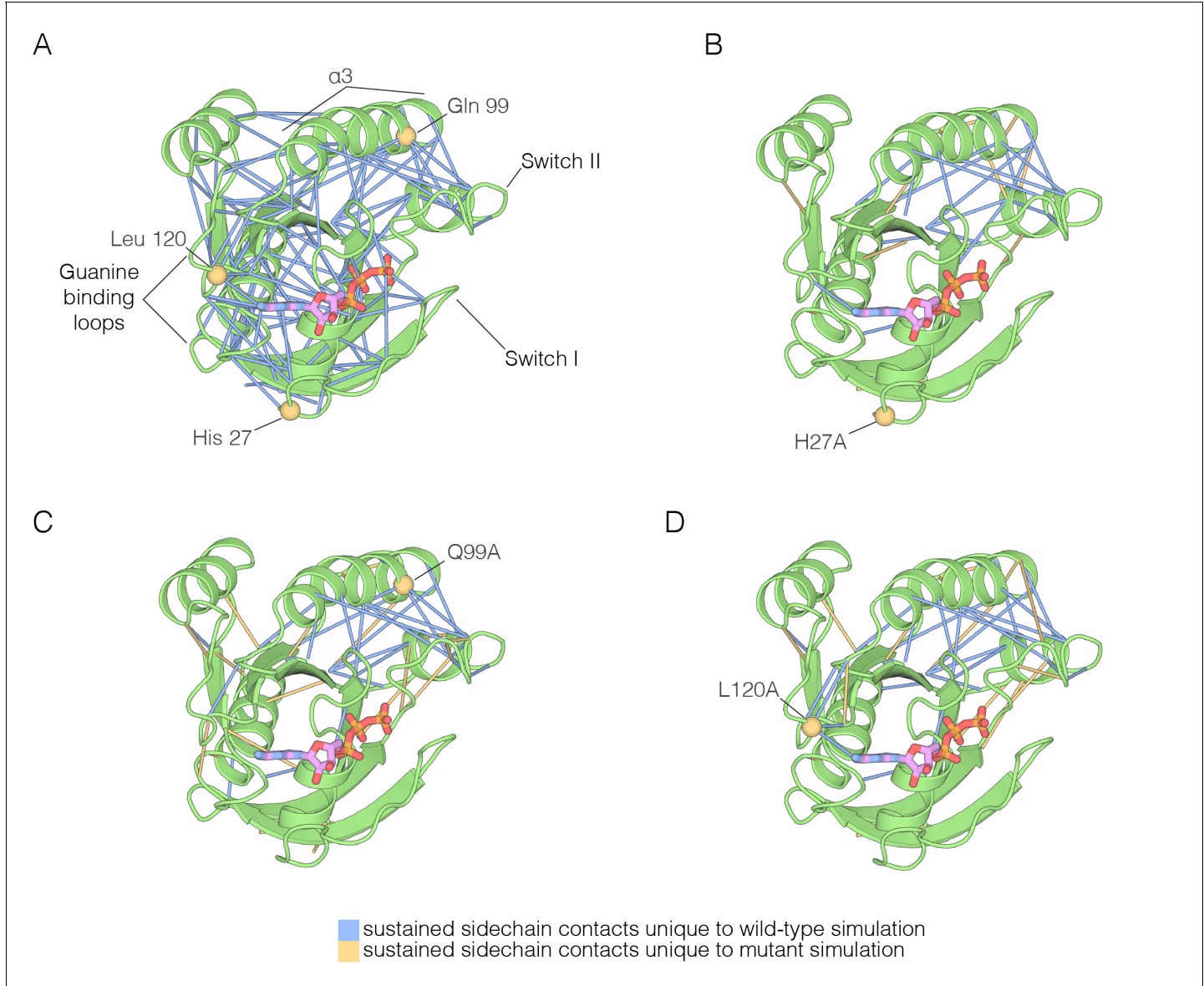

**Figure 7.** The impact of hotspot mutations on structural flexibility in Ras. (**A**) Sustained sidechain-sidechain contacts in the simulation of wild-type Ras•GTP. We used the network analysis tool of Vishveshwara and co-workers (*Bhattacharyya et al., 2013*) to analyze the molecular dynamics trajectories for Ras. Strong interactions between sidechains, as defined by *Bhattacharyya et al. (2013)*, that are present in more than a threshold fraction of 50% of the instantaneous structures in the trajectory are identified by blue lines. These lines are drawn between the Cα atoms of the corresponding residues. (**B**) For simulations of Ras•GTP H27A, strong interactions are shown by blue and yellow lines. The blue lines indicate contacts that are also present in the wild-type simulation. The yellow lines indicate contacts that are unique to the mutant simulation. (**C**). As in (**B**), for Ras•GTP Q99A. (**D**) As in (**B**), for Ras•GTP L120A.

The following figure supplement is available for figure 7:

**Figure supplement 1.** Residue contacts in Ras.

The analysis of stable sidechain contacts shows dramatically different results for the H27A, Q99A and L120A simulations (*Figure 7B–D*). In each of these trajectories, there are far fewer contacts that persist for longer that the 50% threshold we used in the analysis. Switch II and helix α3 remain tightly connected to each other, but the rest of the structure is no longer stably coupled. We emphasize that the loss of stable contacts in the simulations of the mutants does not reflect a general breakdown in the integrity of the structure, but is due instead to a subtle decrease in the maintenance of stable contacts. This can be seen in *Figure 7—figure supplement 1*, which illustrates the interaction between Tyr 137, in helix α4, and His 94, in helix α3. These two residues form a hydrogen-bonding interaction in the crystal structure (*Ting et al., 2015*) (*Figure 7—figure supplement 1A*). In the simulations, His 94 switches between different conformations, but maintains contact with Tyr 137 more frequently in the wild-type simulation than in the L120A simulation (*Figure 7—figure supplement 1B–C*). The simulation results for the Y157A mutant are different than those for the other three. The trajectory for the Y157A mutant does not show a decrease in stable sidechain contacts when compared to the wild-type simulation, although there is evidence for a loosening of the coordination of the nucleotide in both GTP- and GDP-bound simulations (not shown).

In the crystal structure of Ras•GTP, Switch II is positioned such that the sidechain of Gln 61 is not far from the catalytically important conformation that is stabilized by the binding of the GAP. The increased fluctuations seen in Switch II in simulations of mutant Ras tend to move Gln 61 away from catalytically competent conformations, and would therefore be expected to reduce the intrinsic rate of GTP hydrolysis, leading to increased signaling activity. Destabilization of interactions between residues in the guanine binding loops and the guanine base of the nucleotide, or between the P-loop and the phosphate groups of the nucleotide, would increase the intrinsic GDP release rate, leading to activation by the replacement of GDP by GTP. Thus, the simulations support the idea that the hotspot residues analyzed in this study dampen structural fluctuations in wild-type H-Ras that would otherwise disrupt the functional cycle.

## The regulation of Ras is likely to have been altered in the vertebrate lineage

We carried out an evolutionary analysis of Ras proteins by aligning sequences from 72 species and generating an evolutionary tree (*Figure 8A*). This analysis identifies four regions that are the principal sites of divergence between invertebrate and vertebrate Ras sequences, denoted Variable Regions 1 (spanning residues 48 to 52 in human H-Ras), 2 (residues 82 to 101), 3 (residues 121 to 142) and 4 (residues 149 to 155). These variable regions are highlighted in an alignment of the sequence of human H-Ras with Ras from a sponge (*Amphimedon queenslandica*) and from a choanoflagellate (*Salpingoeca rosetta*) (*Richter and King, 2013*) (*Figure 8B*).

In order to capture the essential features that distinguish human H-Ras from invertebrate Ras, we used the evolutionary tree to construct a hypothetical sequence corresponding to the root of the metazoan lineage (*Ashkenazy et al., 2012*). There are 48 differences between the sequences of H-Ras and the hypothetical protein at the base of the metazoan lineage (*Figure 8C*). Of these, 30 represent residues that would activate H-Ras if the wild-type residue were replaced by the residue in the ancestral sequence, based on mutational data in the unregulated-Ras experiment (*Figure 8D*). Eight differences correspond to neutral substitutions, and nine substitutions would decrease function if introduced into H-Ras. We also examined the effects of these substitutions in the data for regulated-Ras in the bacterial two-hybrid experiments. In this case, the trend is the opposite. Eight of the substitutions activate H-Ras, three are neutral, and 36 lead to a decrease in function (not shown). Similar results are obtained by comparing the sequence of human H-Ras to that of extant invertebrate Ras sequences (see *Figure 8—figure supplement 1*, in which the sequence of a choanoflagellate Ras is compared to that of human H-Ras, along with the fitness effects of the substitutions as seen in the unregulated Ras experiment).

To summarize these findings, the ancestral Ras protein contains residues that, when substituted into human H-Ras, are activating in the absence of regulators and detrimental in the presence of a GAP and GEF. This suggests that as the sequence of Ras evolved into the vertebrate lineage, it underwent changes in the mechanisms that prevent unregulated activation, and a co-evolved dependence on the vertebrate GAPs and GEFs. Such co-evolution is consistent with the strong dependence of Ras mutations on the presence or absence of its associated regulatory proteins.

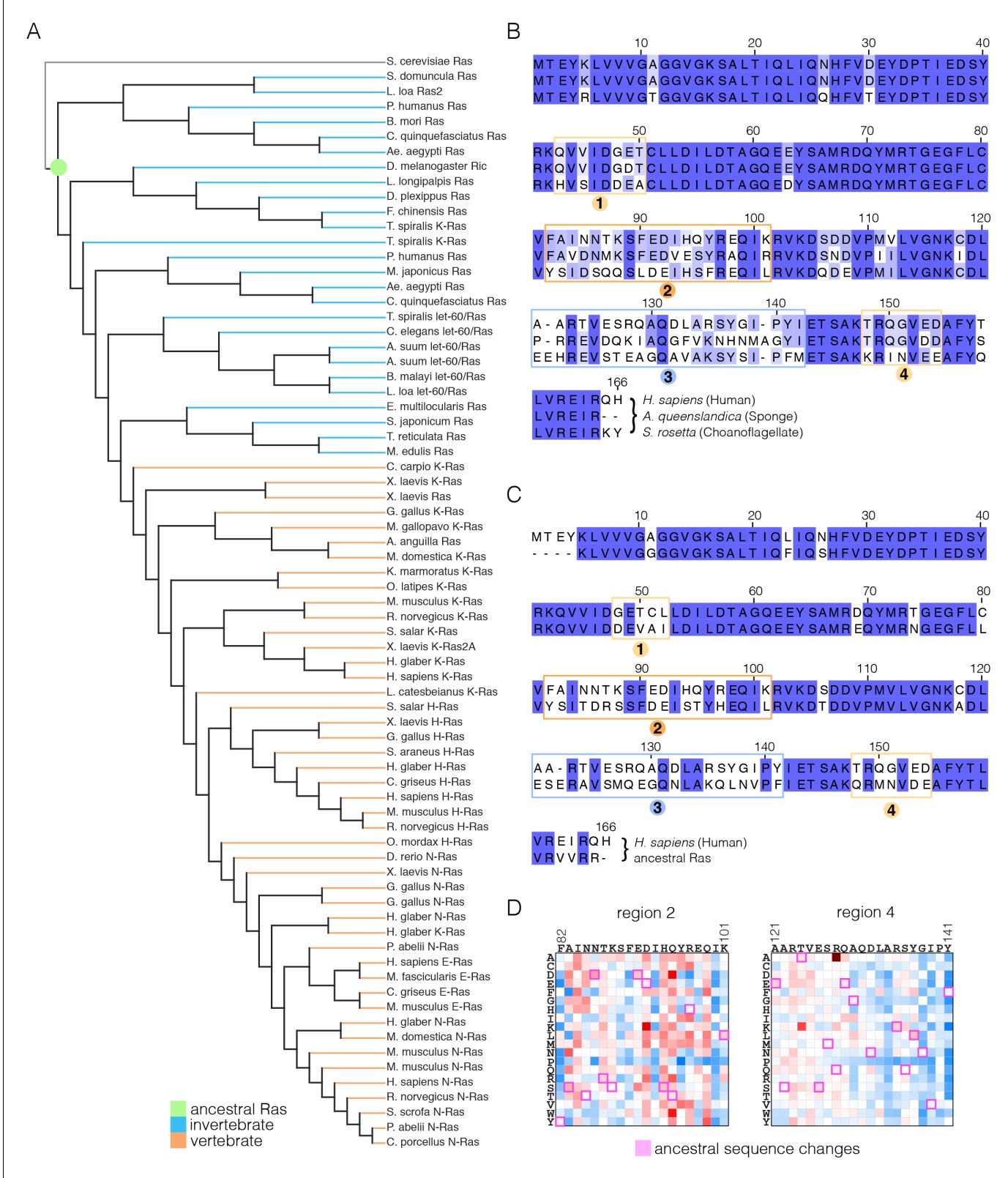

**Figure 8.** Sequence variation in Ras. (**A**) An evolutionary tree from an alignment of 72 extant Ras sequences from invertebrates (blue) and vertebrates (orange). The hypothetical ancestral sequence at the base of the tree is highlighted in green. (**B**) Sequences of Ras from choanoflagellate (*S. rosetta*) and sponge (*A. queenslandica*) are 72% and 80% identical to human H-Ras, respectively. While the sequences are largely identical, there are four principal regions of sequence divergence. These regions correspond to residues 45–50, helix α3, helix α4, and residues 148–154 in human H-Ras. (**C**)
*Figure 8 continued on next page*

*Figure 8 continued*

Comparison of the sequence of human H-Ras to the ancestral sequence from the base of the metazoan lineage reveals similar regions of sequence variation. *S. rosetta* Ras was not used in the alignment of Ras sequences to generate the tree shown in (A). (D) The substitutions in human H-Ras that are present in the ancestral sequence are compared to the mutational data from the unregulated-Ras experiment. There are 48 differences between the sequences of the hypothetical ancestral protein and human H-Ras. Of these, 30 represent residues that would activate unregulated human H-Ras if the wild-type residue were replaced by the residue in the ancestral sequence. Eight differences correspond to neutral substitutions, and nine substitutions would decrease function if introduced into human H-Ras.

The following figure supplement is available for figure 8:

**Figure supplement 1.** Effect of substitutions in human H-Ras that are present in *S. rosetta* Ras.

## The variable regions of Ras are involved in interactions with the Ras activator son-of-sevenless

The Ras-specific nucleotide exchange factor son-of-sevenless (SOS) is unusual in having two binding sites for Ras. One is the canonical binding site that mediates nucleotide exchange (*Boriack-Sjodin et al., 1998*). The other is an allosteric site unique to SOS, which is specific for Ras•GTP, and SOS has low activity unless Ras•GTP is bound at this site (*Margarit et al., 2003*; *Sondermann et al., 2004*) (*Figure 9*). The allosteric binding of Ras•GTP to SOS converts SOS to a highly processive enzyme at the membrane, and this positive feedback loop in Ras activation is necessary for the generation of a sharp response in signaling by the T cell receptor (*Gureasko et al., 2008*; *Das et al., 2009*; *Iversen et al., 2014*). We find that the four Variable Regions correspond to elements in Ras that are correlated with the engagement of SOS at the SOS allosteric site.

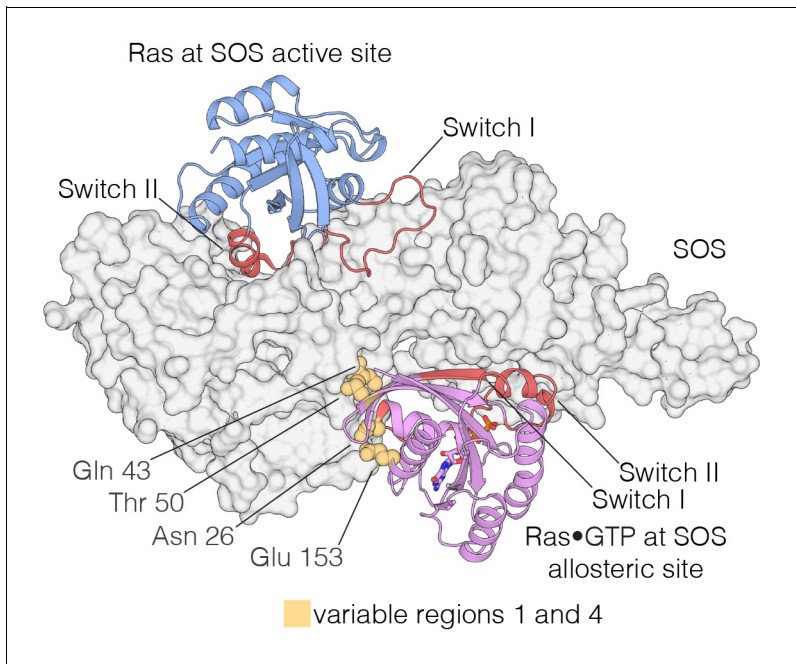

**Figure 9.** Interaction of Ras with the two Ras-binding sites of SOS. Structure of the Ras:SOS complex (PDB code: 1NVV). Two molecules of H-Ras are bound to the allosteric and active site of SOS. The allosteric site of SOS is bound by Ras•GTP, whereas the active site of SOS is bound by nucleotide-free Ras. Switch I and Switch II of Ras (red) are responsible for engaging both sites in SOS. Additionally, residues in variable regions 1 and 4 are involved in the binding of Ras to the allosteric site of SOS, including Asn 26, Gln 43, Thr 50, and Glu 153 (yellow).

The following figure supplement is available for figure 9:

**Figure supplement 1.** SOS-stimulated nucleotide exchange of *S. rosetta* and human Ras.

Variable Regions 1 and 4 in human H-Ras interact directly with SOS at the allosteric site (*Figure 9*), and are altered in *S. rosetta* Ras and in the ancestral sequence (*Figure 8*). We expressed and purified Ras from the choanoflagellate *S. rosetta*. Human SOS stimulates nucleotide exchange with *S. rosetta* Ras and H-Ras with comparable rates, but only when GDP bound to Ras is exchanged for GDP in solution (*Figure 9—figure supplement 1A*). When the exchange reaction is carried out with excess GTP in solution, the reaction shows a marked acceleration in rate for human H-Ras, due to the allosteric stimulation of SOS by Ras•GTP. In contrast, *S. rosetta* Ras is deficient in allosterically stimulating the activity of human SOS (*Figure 9—figure supplement 1B*).

Mattos and colleagues have described alternative conformations of Ras•GTP involving Variable Regions 2 and 3, and also Switch II (*Figure 10A*; the two states are shown schematically in *Figure 10C*) (*Buhrman et al., 2010*; *Holzapfel et al., 2012*). The two conformational states have been denoted 'R' (assumed to be competent for GTP hydrolysis, although this has not been established definitively) and 'T' (assumed to be inactive in terms of GTP hydrolysis). A physiologically relevant allosteric effector that controls the R to T transition has not yet been identified (*Buhrman et al., 2010*; *O'Connor and Kovrigin, 2012*). The two Ras-binding sites in SOS differ in their recognition of the R and T states. SOS binds Ras in the T state conformation at the active site, considering just the conformation of helices α3 and α4, and Switch II (Switch I is displaced during the exchange process) (*Figure 10—figure supplement 1A*). In contrast, SOS recognizes Ras in the R state at the allosteric site (*Figure 10—figure supplement 1B*).

We found that the transition between the R and T states can be triggered by one of the hotspot mutations identified in our study, L120A (*Figure 10B*). We determined the structure of wild-type Ras•GTP and the L120A variant at room temperature, both crystallized with the same crystallographic lattice, and found that wild-type Ras•GTP is in the R state, whereas the L120A mutant is in the T state (there is evidence in the electron density for a minor population of the R state, with an occupancy of 20% or less).

There are no structures for an invertebrate Ras protein in the protein databank. We crystallized Ras from the choanoflagellate *S. rosetta*, and determined structures bound to GDP and GTP at resolutions of 1.85 Å and 1.6 Å, respectively (*Supplementary file 2*). The conformation of Switch II and helix α3 in *S. rosetta* Ras•GTP is very close to that of the T state structure of human Ras (*Figure 10D*), suggesting that the balance between the R and T states of Ras•GTP may be altered in invertebrate Ras with respect to human Ras.

## Concluding remarks

In this paper we have adapted a bacterial two-hybrid system to analyze how mutations affect the functional cycle of human H-Ras. Our strategy is to isolate just the minimal biochemical network that defines this cycle, comprising Ras, its effector Raf, and a GAP and a GEF. Thus, we can analyze the sensitivity of Ras to mutation in the context of this network, while excluding the effects of the membrane and additional regulatory factors. This approach provides an opportunity, for the first time, to use deep mutational scanning approaches to study how local regulatory networks influence the mutational sensitivity and phenotypic plasticity of key signaling molecules.

In the absence of regulators, the distribution of fitness effects of mutations in Ras shows considerable tolerance to mutation, as seen previously in saturation mutagenesis experiments on other proteins (*Soskine and Tawfik, 2010*; *McLaughlin et al., 2012*; *Boucher et al., 2014*; *Podgornaia and Laub, 2015*). However, in the presence of a GAP and a GEF, Ras displays global constraints, where the majority of mutations lead to a modest decrease in function (see *Figure 2*). The data for attenuated-Ras, in the presence of the GAP alone, are similar to that for fully regulated Ras, except that under these conditions Ras is very sensitive to mutations that disrupt GTPase activity, which cause a dramatic gain of function. Taken together, our data show that the local regulatory network places a stringent constraint on the sequence of Ras, and also creates the potential conditions in which it is susceptible to activating mutations. These data extend previous observations that mutational sensitivity in proteins is strongly dependent on the selective conditions in which the protein operates (*Stiffler et al., 2015*). A deeper understanding of the relationship between protein structure and function will emerge as saturation mutagenesis experiments are done in more complete biological contexts.

The structural fold of Ras is intrinsically capable of accommodating sequence changes that lead to the acquisition of new function. For example, paralogous members of the Ras superfamily show a

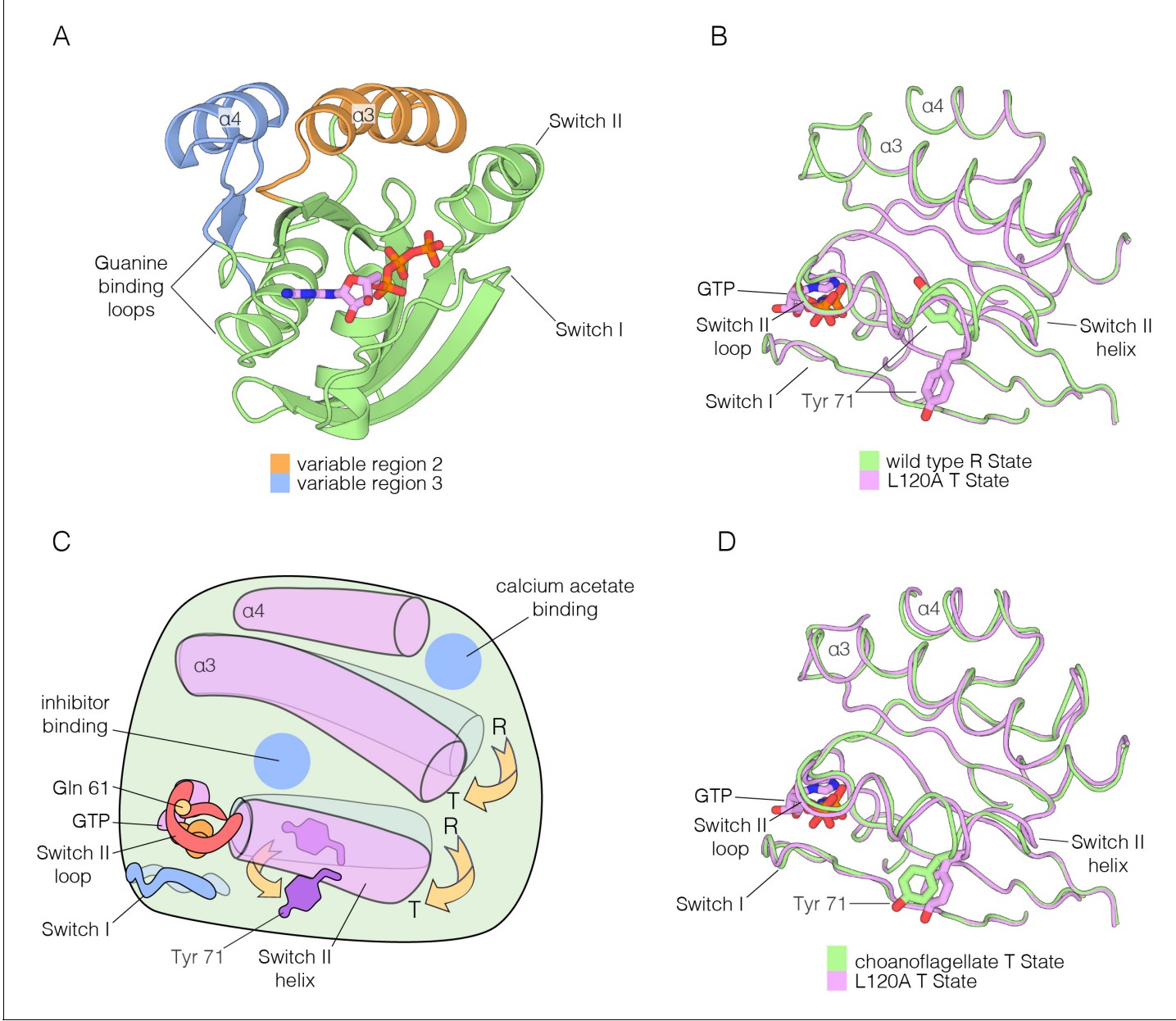

**Figure 10.** Comparison of the R and T states in Ras. (**A**) Variable Regions 2 and 3 highlighted on the structure of Ras•GTP. Variable Region 2 comprises helix α3 and the preceding loop, and Variable Region 3 comprises helix α4 and the preceding loop that is partially involved in guanine binding. (**B**) Room-temperature crystal structure of wild-type Ras adopts the R state as defined by Mattos and colleagues (***Buhrman et al., 2010***; ***Holzapfel et al., 2012***), where the sidechain of Tyr 71 is buried in the hydrophobic core of the protein. Crystal structure of the L120A mutant shows a rotation of Tyr 71, whereby the sidechain is exposed. Additionally, the interface between helix α3 and Switch II undergoes a conformational change. (**C**) Schematic of the conformational transition between the R state and T state as defined by Mattos and colleagues (***Buhrman et al., 2010***; ***Holzapfel et al., 2012***). Helix α3 and the C-terminal helix of Switch II shift downwards in the transition from the R state to the T state, leading to a rotation of Tyr 71 from a buried to an exposed conformation. The blue circles indicate binding sites for calcium acetate (Buhrman et al., 2010) and small molecule inhibitors. (**D**) Comparison of the choanoflagellate *S. rosetta* crystal structure to the L120A structure. *S. rosetta* Ras adopts the T state, similar to L120A, where Tyr 71 is rotated outwards and exposed.

The following figure supplement is available for figure 10:

**Figure supplement 1.** R and T states of Ras bound to SOS.

high degree of structural conservation within the core elements of the GTPase domain despite considerable sequence divergence (*Rojas et al., 2012*). In contrast, orthologous members of the family (such as the Ras proteins) show high sequence conservation across metazoan evolution. We propose that a critical aspect of function of these proteins is the tuning of the GTPase cycle such that switching between active and inactive states and responsiveness to regulators is matched to its specific biological context. More generally, we suggest that high conservation in orthologs of a protein family may arise from the pressures imposed by the dynamics of local biochemical environments.

Ras itself has undergone a further specialization in the transition from invertebrates to vertebrates. Our analysis indicates that this specialization is associated with the acquisition, or modification, of allosteric control, as indicated by the fact that the majority of these sequence changes involve residues that lead to activation when mutated in human Ras. One such control mechanism is the ability of Ras to allosterically activate its own activator, SOS, a property that may have emerged in higher vertebrates. Additional regulation, involving interactions at the membrane (*Abankwa et al., 2008*; *Mazhab-Jafari et al., 2015*), the potential dimerization of Ras (*Muratcioglu et al., 2015*), and interaction with different effector proteins, may also play a role in the emergence of the new levels of allosteric control.

Oncogenic mutations that disturb the switching mechanisms of Ras result in aberrant signaling and cancer, highlighted by the fact that Ras is one of the most important proto-oncogenes in the human genome (*Young et al., 2009*; *Prior et al., 2012*). An important result from our experiments that merits further study is that oncogenic mutations, such as G12V, are conditionally neutral (see *Figure 2*). That is, such mutations are minimally disruptive in the presence of the regulators, which may allow these mutations to appear and float in the population, manifesting a dramatic gain of function only upon perturbation to the regulators of the Ras switching cycle (see *Figures 3* and *4*). Thus, normal physiology may provide evolutionary paths to acquiring neutral mutations that, with a second hit such as loss of regulation, can display pathological activities. Such mutations could conceptually be analogous to 'latent driver' mutations that emerge prior to and during tumor evolution (*Nussinov and Tsai, 2015*).

The extensive experience with cancer drugs that target protein kinases has shown that resistance to these drugs emerges quickly, often through mutation at sites of allosteric communication with the drug-binding site (*Shah et al., 2002*; *Azam et al., 2003*; *Branford et al., 2009*). Small molecule inhibitors of Ras have yet to achieve clinical relevance, but there is a concerted effort underway to obtain such inhibitors (*McCormick, 2016*). Experience with kinase inhibitors suggests that the clinical introduction of Ras inhibitors will be followed by the rapid acquisition of resistance mutations in patients, necessitating further cycles of inhibitor development to overcome resistance. Our analysis of the mutational sensitivity of Ras-G12V has indicated that there are epistatic interactions that reduce the sensitivity to mutation of other elements of the structure, such as Switch II. The mapping of such epistatic interactions for other oncogenic mutations will provide an important conceptual framework for understanding the response of Ras to inhibition by small molecules.

## Materials and methods

### Bacterial two-hybrid selection assay

We utilized a modified version of a bacterial two-hybrid system in which transcription of chloramphenicol acetyltransferase (pZERM1-CAT plasmid, pRM+ promoter, ampicillin resistant) is made dependent on the binding between each human H-Ras variant (fused to the N-terminal domain of the *E. coli* RNA polymerase α-subunit on the pZA31 plasmid, doxycycline inducible, kanamycin resistant) and the RBD from human C-Raf kinase (fused to the bacteriophage λ-cI DNA binding domain on the pZS22 plasmid, IPTG inducible, trimethoprim resistant). p120 RasGAP was expressed from the pZS22 plasmid, separated from the Raf-RBD by an internal ribosome binding site, and RasGRP1 was expressed from the pZERM1 plasmid on an IPTG-inducible cassette independent of the reporter CAT gene.

Electrocompetent MC4100-Z1 cells containing pZERM1-CAT and pZS22 plasmids were transformed with 100 ng of the pZA31-RNA polymerase α-Ras variant library, recovered for 2 hr in LB media, and grown to saturation overnight in LB media containing 20 μg/mL trimethoprim, 50 μg/mL kanamycin, and 100 μg/mL ampicillin. 20 μL of saturated culture was diluted to an optical density

(OD) at 600 nm of 0.001 for a 2 hr outgrowth in 30 mL LB at the same antibiotic concentrations. Cells were diluted to an OD of 0.0001 and induced with 50 ng/mL doxycycline and 100 μM IPTG for 3 hr in 60 mL of LB + antibiotics. After induction, ~50 mL of culture was reserved for deep-sequencing of the pre-selection population. Selection cultures were started with the remainder of the induction cultures at an OD of 0.001 in 60 mL of LB + antibiotics + inducer with 75 μg/mL chloramphenicol for 7 hr, ensuring that the OD remained below 0.1 over the course of the experiment. Both pre- and post-selection cultures were subject to plasmid DNA isolation, PCR amplification of the Ras variant library, and standard preparation for Illumina MiSeq paired-end sequencing (*Bentley et al., 2008*).

## Construction of the Ras variant libraries

Comprehensive single-mutant libraries were constructed using oligonucleotide-directed mutagenesis of human H-Ras. To mutate each residue in wild-type and G12V H-Ras (residues 2–166), two mutagenic oligonucleotides (one sense, one antisense) were synthesized (Integrated DNA Technologies), with sequence complementary to 15 base pairs on either side of the targeted codon. For the targeted codon, the oligonucleotides contain NNS codons, in which an N is a mixture of A, T, G, and C and S is a mixture of G and C. This biased randomization results in 32 possible codons with all 20 amino acids sampled – a significant decrease in library complexity, and a significant reduction in stop-codon frequency, without loss of amino acid complexity (*McLaughlin et al., 2012*; *Raman et al., 2016*). One round of PCR was carried out with and an antisense or sense oligonucleotide flanking the Ras gene. A second round of PCR was carried out using a combination of the first-round products and both flanking primers, and produced the full-length double-stranded product containing a single degenerate codon. For the 165 randomized positions, we carried out 165*2 first-round reactions and 165 second-round reactions, for a total of 495 reactions. All reactions yielded single intense bands on agarose gels, and these products were purified using standard gel extraction protocols. PCR product concentrations were measured using Picogreen (ThermoFisher), pooled in an equimolar ratio, gel purified, digested with Bsu36I and XhoI, and ligated with T4 DNA ligase overnight with the manufacturer's recommended protocol into the pZA31 expression vector. The ligation product was purified, eluted into 10 μL ddH2O, and transformed into 100 μL electrocompetent TOP10 *E. coli* cells (Thermo Fisher). We limited our Illumina sequencing to the use of 300 base paired-end reads, rather than longer reads, to ensure high-quality base calls. Since these read lengths cannot cover the full Ras gene, our scanning mutagenesis library was generated as three separate sub-libraries, covering mutations at residues 2–56, 57–111, and 112–166. Each sub-library was independently subject to the bacterial two-hybrid assay, amplification, and sample preparation for MiSeq sequencing. Samples from each sub-library were uniquely barcoded so that all samples from a single experiment were analyzed on a single MiSeq chip.

## Protein expression and purification

The *S. rosetta* cDNA library was a kind gift from Nicole King (*Fairclough et al., 2013*). The coding region for the catalytic domain of *S. rosetta* Ras residues 1–167 was amplified from the cDNA library and cloned into pProEX HTb expression vector using the NcoI – XhoI restriction sites. Human H-Ras variants were cloned into a pET-28b expression vector modified with a 5' His6-tag and a TEV protease site coding sequence, and variants were generated by Quikchange site-directed mutagenesis (Agilent).

Hexahistidine-tagged recombinant human H-Ras variants (residues 1–166) were transformed into 50 μL *Escherichia coli* (BL21 (DE3)) in the pET-28b expression vector. After bacterial growth to an OD of 0.5 in Terrific Broth containing 50 mg/mL kanamycin at 37°C, induction was carried out at 18°C with 0.5 mM IPTG, and growth was continued at 18°C for 18 hr. The bacteria were pelleted by centrifugation and the pellet was either stored at −80°C or used freshly for subsequent purification steps.

The pellet was resuspended in lysis buffer (50 mM Tris, 500 mM NaCl, 20 mM imidazole, 5% glycerol, pH 8.0) containing protease inhibitor cocktail (Roche) and 2 mM $\beta$-mercaptoethanol (BME). The cells were lysed by sonication, and the cell debris was removed by centrifugation. The supernatant was applied to a 5 mL nickel column (HisTrap FF, GE Healthcare) and the column was washed with 50 mL lysis buffer and 50 mL low salt buffer (50 mM Tris, 50 mM NaCl, 20 mM imidazole, 5% glycerol, pH 8.0). The nickel column was then connected directly to an ion-exchange chromatography

column (HiTrap Q FF, GE Healthcare) and the protein was eluted with elution buffer (50 mM Tris, 50 mM NaCl, 250 mM imidazole, 5% glycerol, pH 8.0). The protein was then eluted from the ion-exchange column in a salt gradient from 50 mM to 1M NaCl to give the partially purified protein, usually in the following buffer: 50 mM Tris, ~230 mM NaCl, 5% glycerol, pH 8.0. 100 µM GDP and 1 mM TCEP was added to the protein solution and the hexahistidine tag was then cleaved overnight using hexahistidine-tagged TEV protease (1 mg TEV per 25 mg crude Ras). The protein solution was run over a nickel column and washed with 35 mL low salt buffer to separate the protein from the cleaved hexahistidine tag and TEV protease. The protein was concentrated to ~40 mg/mL and either frozen at −80°C or immediately used for GMP-PNP loading. The purification protocol for *S. rosetta* Ras was identical to that of human H-Ras.

Ras was loaded with GMP-PNP using a protocol modified from Eberth and Ahmadian (*Eberth and Ahmadian, 2009*). A tenfold molar excess of GMP-PNP was added to ~10 mg crude Ras and bound nucleotide was cleaved by 10 U bovine alkaline phosphatase (Sigma) and 1 mM $ZnCl_2$ at 25°C for 1 hr. Free nucleotide and alkaline phosphatase were removed by size exclusion chromatography using a Superdex 75 column (10/300 GL, GE Healthcare) in the following buffer: 40 mM HEPES, 150 mM NaCl, 4 mM $MgCl_2$, 1 mM TCEP, 5% glycerol, pH 7.4. The protein was concentrated to ~40 mg/mL for further use.

The purification protocol for human p120 RasGAP was identical to that for Ras, except during the ion-exchange chromatography step, where the protein was eluted in a salt gradient from 10 to 250 mM NaCl. The purification procedure for human RasGRP1 was also similar to that of Ras, except during the size exclusion chromatography step, where the protein was purified using a Superdex 200 column (26/600, GE Healthcare).

Hexahistidine-tagged, SUMO-fused recombinant human C-Raf-RBD (residues 55–131) was transformed into *Escherichia coli* (BL21 (DE3)) in the pET-SUMO expression vector. Cells were grown to an OD of 0.5 in Terrific Broth containing 50 mg/mL kanamycin at 37°C and induction was carried out at 37°C with 0.5 mM IPTG for 7 hr. Cells were lysed by sonication and affinity purified on a nickel column, using a protocol similar to that for Ras. Protein was eluted from the nickel column in the following elution buffer: 50 mM Tris, 50 mM NaCl, 250 mM imidazole, 5% glycerol, pH 7.5. The protein was immediately desalted into elution buffer without imidazole and SUMO was removed by the addition of ULP1 protease (Thermo Fisher) overnight with 1 mM TCEP. The protein was run over a nickel column to remove hexahistidine-tagged SUMO and the cleaved protein was bound to an ion-exchange chromatography column (HiTrap SP HP, GE Healthcare). The protein was eluted in a salt gradient from 50 to 1 M NaCl and Raf-RBD eluted at 300 mM NaCl. The Raf-RBD was further purified by size exclusion chromatography, concentrated, and stored in the following buffer: 40 mM HEPES, 150 mM NaCl, 1 mM TCEP, 5% glycerol, pH 7.4.

## Determining nucleotide loading state of Ras variants by HPLC

HPLC was used to determine the loading state of Ras variants after loading with GMP-PNP. A reverse-phase C18 analytical column (Targa) and a precolumn were equilibrated in a buffer containing 100 mM $KH_2PO_4$, 10 mM tetrabutylammonium bromide, and 7.5% acetonitrile. A mixture of pure nucleotides, containing GMP, GDP, GTP, and GMP-PNP was first injected as a reference to determine the elution time of each species. Ras variants after GMP-PNP loading were next injected and bound nucleotide retention time was compared to the reference injection to verify that the variants were loaded with GMP-PNP.

## Crystallization and structure determination of Ras variants

Crystals of wild-type human H-Ras bound to GMP-PNP were grown by mixing 1 µL of protein solution (17 mg/mL) with 1 µL of reservoir buffer (16% PEG3500, 100 mM calcium acetate, pH 7.5). Human H-Ras mutant L120A bound to GMP-PNP was crystallized by mixing 1 µL of protein solution (40 mg/mL) with 1 µL of reservoir buffer (17% PEG8000, 200 mM calcium acetate, 100 mM MES/NaOH, pH 6.0). Drops were allowed to equilibrate in a sitting drop tray over 500 µL reservoir solution for one week at 20°C.

Crystals of *S. rosetta* Ras with GDP were obtained by mixing 2 µL of 30 mg/mL protein with 2 µL of the reservoir buffer (20% PEG3350, 100 mM Tris-HCl pH9.0, 100 mM $CaCl_2$) at 20°C. Crystals

were cryo-protected by adding the cryo-protection buffer (20% PEG3350, 100 mM Tris-HCl pH 9.0, 100 mM $CaCl_2$, 15% glycerol) in one step, and flash frozen in liquid nitrogen.

The GMP-PNP bound *S. rosetta* Ras was crystallized by mixing 1 µL of 18 mg/mL protein with 1 µL of the reservoir buffer (15% PEG8000, 100 mM Bis-Tris pH 5.5, 200 mM magnesium acetate, 50 mM $CaCl_2$) at 20°C. The best crystals were obtained by streak-seeding. Crystals were added the cryo-protection buffer containing (15% PEG8000, 100 mM Bis-Tris pH 5.5, 200 mM magnesium acetate, 50 mM $CaCl_2$, 12.5% ethylene glycol) before being flash-frozen in liquid nitrogen.

Preliminary diffraction data for room-temperature wild-type human H-Ras were collected at the Stanford Synchrotron Radiation Lightsource, beamline 9–2. Final diffraction data presented herein were collected at the Advanced Light Source (ALS), beamline 8.3.1. For low temperature data, crystals were cryo-protected with reservoir buffer supplemented with 20% glycerol and flash cooled with liquid nitrogen. Low temperature data were collected at 100 K on ALS beamline 8.2.1. For room temperature data, crystals were looped directly from the sitting drop tray into polyimide mounts (MiTeGen MicroMounts), covered with a polyester sleeve containing a small amount of reservoir solution in the tip, and immediately placed in a cryostream at 277 K and irradiated with a defocused beam of 14 keV X-rays. Data were integrated with XDS (*Kabsch, 2010*) and scaled and reduced to the asymmetric unit with Aimless in the CCP4 suite (*Winn et al., 2011*). Initial structures for both wild-type and mutant Ras were produced with phases from a structure of calcium acetate-bound Ras in the same space group (PDB code 3K8Y; *Buhrman et al., 2010*). The model was rebuilt using Coot (*Emsley and Cowtan, 2004*) and iteratively refined with Phenix (*Adams et al., 2010*). These programs were provided through SBGRID (*Morin et al., 2013*). Positive difference electron density around the modeled magnesium ion associated with bound GMP-PNP was observed in both L120A structures. Considerable anomalous signal was also observed at this position. The ion position was therefore refined using Phenix with both metals present, with variable occupancy. The observed density was best accounted for with 70% calcium ion and 30% magnesium ion.

Diffraction data for *S. rosetta* Ras were collected at ALS beamline 8.2.1 and 8.3.1. Data were integrated in XDS (*Kabsch, 2010*) and scaled with Scala (*Evans, 2006*) or Aimless. For GDP-bound *S. rosetta* Ras, a molecular replacement solution was found in Phaser (*McCoy, 2007*) using the H-Ras structure (PDB ID: 5P21; [*Pai et al., 1990*]). The structural model was rebuilt using Coot (*Emsley and Cowtan, 2004*) and refined by Refmac (*Murshudov et al., 2011*) and Phenix. The model was used to find a molecular replacement solution for GMP-PNP-bound *S. rosetta* Ras. The data were processed in space group $C222_1$ and molecular replacement was successful, but the model could not be refined. We used Zanuda (*Lebedev and Isupov, 2014*) to help resolve the space group as $P2_1$. The crystal was pseudo-merohedrally twinned (twin fraction estimated at 0.47) (*Hamdane et al., 2009*). The structure was successfully rebuilt and refined in $P2_1$ with Coot and Phenix, without accounting for the apparent twinning. *Supplementary file 1* reports the final model R-factors calculated in Phenix without accounting for twinning and for the same model (no further refinement) calculated with Refmac using the twin operator l, -k, h.

The structures were deposited in the Protein Data Bank with the following PDB codes: wild-type H-Ras at 277K: 5WDO; H-Ras L120A at 277K: 5WDP; H-Ras L120A at 100K: 5WDQ; *S. rosetta* Ras with GMP-PNP: 5WDR; *S. rosetta* Ras with GDP: 5WDS.

## Raf-RBD binding assays

Ras•GTP:Raf-RBD binding assays were performed on a MicroCal-autoITC 200 instrument (GE Healthcare). GMP-PNP loaded Ras and Raf-RBD samples were diluted in a binding buffer containing 40 mM HEPES, 150 mM NaCl, 4 mM $MgCl_2$, 1 mM TCEP, 5% glycerol, pH 7.4. Final Ras and Raf-RBD concentrations were typically 100 µM and 10 µM, respectively. Each set of ITC experiments included three samples: 420 µL of Raf-RBD in the cell, 140 µL Ras in the syringe, and 400 µL of binding buffer. All samples were stored at 4°C before the titration experiments.

The ITC experiments were performed at 25°C. An initial injection of 0.5 µL was excluded from data analysis, followed by 15 injections of 2.6 µL each, separated by 420 s with a filter period of 5 s. The protein solution was stirred at 500 rpm over the course of the titration. Titration curves were fit with a one-site binding model, where the three fitting parameters are stoichiometry, *N*, association constant, $K_A$, and binding enthalpy, *ΔH*. The binding entropy is then calculated using the formula: $\Delta S = (\Delta H + RT*ln(K_A))/T$.

## In vitro GTP hydrolysis assay

GTP hydrolysis assays were performed with recombinant *E. coli* phosphate-binding protein labeled with the MDCC fluorophore (phosphate sensor, Thermo Fisher). First, each Ras mutant was loaded with GTP by incubating protein with a ten-fold molar excess of EDTA in the presence of a ten-fold molar excess of GTP. The loading reaction was performed on ice for at least one hour and the reaction was quenched by the addition of a twenty-fold molar excess of $MgCl_2$. The reaction was then buffer exchanged using a small scale NAP-5 column (GE Healthcare) in order to remove excess nucleotide and salt.

For the GTP hydrolysis assay, 1 µM Ras was mixed with 0.5 µM phosphate sensor and 0.25 µM p120 RasGAP in the following buffer: 40 mM HEPES, 150 mM NaCl, 4 mM $MgCl_2$, 1 mM TCEP, 5% glycerol. The reaction progress curve was monitored on a Tecan fluorescence plate reader with measurements taken every 15 s. Data were fit to a single exponential for quantification of GAP-stimulated GTP hydrolysis rates.

## In vitro nucleotide exchange assay

GDP release assays were performed with Ras bound to mant-dGDP (Axxora Biosciences), loaded with the protocol described previously. 1.5 µM Ras was mixed with 2.5 µM GEF and 3.5 mM GDP in solution in the following buffer: 40 mM HEPES, 150 mM NaCl, 4 mM $MgCl_2$, 1 mM TCEP, 5% glycerol. Nucleotide exchange rates were monitored on a Tecan fluorescence plate reader with measurements taken every 15 s. Data were fit to a single exponential decay curve for quantification of GEF-mediated GDP exchange rates.

## Yeast Ras growth assay

Human H-Ras was cloned into the p416-TEF plasmid using gap-repair cloning in JRy4012 *S. cerevisiae* yeast cells kindly provided by Jasper Rine (*Joska et al., 2014*). Plasmids containing Ras or empty vector were transformed into the same yeast strain on SD –Ura plates, and individual colonies were picked and used to inoculate a 5 mL overnight culture in SD –Ura broth at 30°C. The overnight culture was used to start a new culture in 5 mL YPD medium, and cells were allowed to grow to an OD of ~0.8 at 30°C. Once the cells reached the desired OD, 1 mL of each culture was subject to heat shock at 55°C for 5 min, as described (*Sass et al., 1986*). Five-fold serial dilutions of the heat shock cultures were plated onto SD –Ura plates and were grown at 30°C for two days and colonies were subsequently imaged.

## Molecular dynamics simulations

Simulations of Ras in this study were carried out using the AMBER force field (*Case et al., 2014*) combined with the ff99SB-ILBN backbone correction (*Lindorff-Larsen et al., 2010*) and the TIP3P water model (*Mahoney and Jorgensen, 2000*). The simulated systems were solvated in a water box with a minimum of 12.5 Å separation from any protein atom and the box edges, and residue protonation states corresponded to pH 7. The protein backbone atoms were restrained to their initial positions using a harmonic potential with a force constant of 1 kcal mol$^{-1}$ Å$^2$ for 5 ns as an equilibration step and restraints were subsequently removed. Simulations were performed in the NPT ensemble with T = 300 K and p=1 bar. Water molecules and all bond lengths were constrained using M-SHAKE (*Kräutler et al., 2001*). Long-range electrostatic energies were calculated using particle-mesh Ewald summation. The simulation time step was 2 fs. Network analysis of sidechain-sidechain interactions was carried out using the PSN-Ensemble software (*Bhattacharyya et al., 2013*).

## Ancestral sequence reconstruction

We obtained an initial set of 21,243 Ras superfamily (PF00071) sequences from the Pfam protein families database (*Finn et al., 2016*). Out of the 8988 metazoan sequences in the alignment, we selected a set of 72 Ras sequences for evolutionary analysis, with yeast Ras added as an outgroup. The corresponding maximum likelihood phylogenetic tree and sequence alignment were jointly inferred with PASTA (*Mirarab et al., 2015*). We used four iterative alignment and phylogenetic inference steps. RAxML (*Stamatakis, 2014*) was used as the tree estimator with the PROTCATLGF model, which employs the LG amino acid substitution matrix (*Le and Gascuel, 2008*) and optimizes site-specific substitution rates. PASTA likelihood scores indicated convergence of the alignment.

Ancestral sequences were then reconstructed via maximum likelihood with FastML (*Ashkenazy et al., 2012*) using the same substitution matrix, with the phylogenetic tree and alignment from PASTA as input. Most residues in the hypothetical metazoan ancestral sequence were reconstructed by FastML with high confidence (median probability of reconstructed amino acids was 97%; only 10 of 178 residues had probabilities below 50%).

We also tested our results on the effect of substitutions in the reconstructed ancestral Ras sequence for robustness to uncertainty in the reconstructed sequence. Averaging over the probability of each reconstructed residue at each site, we found that the average number of substitutions in the ancestral sequence relative to human H-Ras was 48.5 (compared with 48 for the most likely sequence). The average number of substitutions that would increase/have little effect on/decrease unregulated Ras function was 30.5/9.0/9.0, respectively, in good agreement with 30/8/9 for the most likely sequence. Similarly, for regulated Ras the average numbers were 9.2/2.9/36.4, compared with 8/3/36 for the most likely sequence. Locations of likely substitutions were also consistent with the patterns displayed in the most likely ancestral sequence shown in *Figure 8C*. Thus, we find that uncertainty in the amino acid composition of the reconstructed ancestral sequence does not affect our findings about the predicted effects of substitutions relative to the human H-Ras sequence.

## Acknowledgements

We thank William Russ and Arjun Raman for assistance with the bacterial two-hybrid system, library construction, and deep sequencing; Xiaoxian Cao for assistance with cloning; scientists at ALS beamlines 8.2.1, 8.2.2, and 8.3.1 and SSRL 9–2 for help with data collection; Nicole King for the kind gift of the *S. rosetta* cDNA library; and members of the Kuriyan, Ranganathan and Kortemme labs for helpful discussions. This work was supported in part by NIH grant PO1 AI091580 to AKC and JK. RR acknowledges support from the Robert A. Welch Foundation (I-1366), the Lyda Hill Endowment for Systems Biology, and the Green Center for Systems Biology. NHS is supported by the Damon Runyon Cancer Research Foundation postdoctoral fellowship.

## Additional information

### Competing interests

JK: Senior editor, *eLife*. AKC: Senior editor, *eLife*. The other authors declare that no competing interests exist.

### Funding

| Funder | Author |
| --- | --- |
| Howard Hughes Medical Institute | Pradeep Bandaru<br>Neel H Shah<br>Moitrayee Bhattacharyya<br>Joshua C Cofsky<br>Christine L Gee |
| Damon Runyon Cancer Research Foundation | Neel H Shah |
| National Institutes of Health | John P Barton<br>Arup K Chakraborty<br>Rama Ranganathan<br>Yasushi Kondo |

The funders had no role in study design, data collection and interpretation, or the decision to submit the work for publication.

### Author contributions

PB, Conceptualization, Data curation, Software, Formal analysis, Validation, Investigation, Visualization, Methodology, Writing—original draft, Writing—review and editing; NHS, Conceptualization, Writing—review and editing; MB, Software; JPB, Software, Formal analysis; YK, JCC, CLG, Data curation; AKC, Formal analysis; TK, Conceptualization, Supervision, Writing—review and editing; RR, Conceptualization, Resources, Supervision, Funding acquisition, Investigation, Writing—original

draft, Project administration, Writing—review and editing; JK, Conceptualization, Resources, Data curation, Software, Formal analysis, Supervision, Funding acquisition, Validation, Investigation, Visualization, Methodology, Writing—original draft, Project administration, Writing—review and editing

### Author ORCIDs

Pradeep Bandaru, (iD) http://orcid.org/0000-0002-9354-3340
Neel H Shah, (iD) http://orcid.org/0000-0002-1186-0626
John P Barton, (iD) http://orcid.org/0000-0003-1467-421X
Christine L Gee, (iD) http://orcid.org/0000-0002-2632-6418
John Kuriyan, (iD) http://orcid.org/0000-0002-4414-5477

## Additional files

### Supplementary files

• Supplementary file 1. Raw fitness data from bacterial two-hybrid screen.

• Supplementary file 2. Crystallographic data collection and refinement statistics.

### Major datasets

The following datasets were generated:

| Author(s) | Year | Dataset title | Dataset URL | Database, license, and accessibility information |
|---|---|---|---|---|
| Joshua C Cofsky, Pradeep Bandaru, Christine L Gee, John Kuriyan | 2017 | H-Ras bound to GMP-PNP at 277K | http://www.rcsb.org/pdb/explore/explore.do?structureId=5WDO | Publicly available at the RCSB Protein Data Bank (accession no. 5WDO) |
| Joshua C Cofsky, Pradeep Bandaru, Christine L Gee, John Kuriyan | 2017 | H-Ras mutant L120A bound to GMP-PNP at 277K | http://www.rcsb.org/pdb/explore/explore.do?structureId=5WDP | Publicly available at the RCSB Protein Data Bank (accession no. 5WDP) |
| Pradeep Bandaru, Christine L Gee, John Kuriyan | 2017 | H-Ras mutant L120A bound to GMP-PNP at 100K | http://www.rcsb.org/pdb/explore/explore.do?structureId=5WDQ | Publicly available at the RCSB Protein Data Bank (accession no. 5WDQ) |
| Yasushi Kondo, Christine L Gee, John Kuriyan | 2017 | Choanoflagellate Salpingoeca rosetta Ras with GMP-PNP | http://www.rcsb.org/pdb/explore/explore.do?structureId=5WDR | Publicly available at the RCSB Protein Data Bank (accession no. 5WDR) |
| Yasushi Kondo, Christine L Gee, John Kuriyan | 2017 | Choanoflagellate Salpingoeca rosetta Ras with GDP bound | http://www.rcsb.org/pdb/explore/explore.do?structureId=5WDS | Publicly available at the RCSB Protein Data Bank (accession no. 5WDS) |

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
