## [Decision Letter]

Thank you for submitting your article "Deconstruction of the Ras switching cycle through saturation mutagenesis" for consideration by *eLife*. Your article has been favorably evaluated by Philip Cole (Senior Editor) and four reviewers, one of whom, Alfonso Valencia (Reviewer #1), is a member of our Board of Reviewing Editors.

This work presents a remarkable tour-de-force aiming at the roles of every residue in the Ras switching cycle through saturation mutagenesis.

Human wild-type Ras and mutants were expressed alone ("unregulated-Ras") and in the presence of the GAP and the GEF ("regulated-Ras") or only with GAP ("attenuated-Ras"). Additional experiments were carried out with proteins that contain the oncogenic mutation Ras-G12V ("Ras-G12V"). The experimental system is a variant of the yeast two-hybrid system in which the Ras proteins are coupled to CAT production via the interaction with a Raf-RBD. Function was quantified in terms of cell growth during selection with chloramphenicol.

The outcome is what we would intuitively expect: the allosteric switching function – particularly of a key hub in the cellular network like Ras – restricts the sequence. Most mutations mildly reduce the function of Ras. More severe effects are at residues that are functionally critical, such as binding to GTP, to Raf-RBD and to regulators.

The paper provides a large body of data on the activity of the variants obtained by mutation scanning. All this information is accompanied with a detailed structural interpretation, and complemented with a new X-ray structure and MD simulations.

This comprehensive analysis provides an anatomical sequence-structure-function map of a protein that controls critical cellular proliferation pathways. In summary, the data is very interesting and valuable.

Unfortunately, the interpretation of the data left many open questions. Indeed, the overall goal of the paper, beyond the detailed description of some of the mutants, is not completely clear.

The relations between the proposed hot spots, the interactions with effectors and the molecular details of the allosteric mechanism, are not addressed clearly in the paper and the relation of the effectors with the intrinsic allosteric mechanism will have to be described explicitly.

The arguments around the possible evolutionary path to the acquisition of the GTPase regulation in vertebrates are not very strong, and the underlying analysis of the sequences does not seem to be at the same quality level than the rest of the paper. The direct relation between evolutionary analysis and the allosteric control in Ras should be clarified.

Other more specific comments include:

Terminology, like 'latchkey' which is not well defined and 'allostery' quite loosely. The 'latch-like' interaction terminology is also confusing.

The protein names, i.e. Ras, Raf and Rap are used as general terms, without reference to isoforms. Isoforms of Ras, Rap and Raf differ in sequence and have distinct signaling properties, so it is unclear why the investigators do not address isoform differences.

Consider adopting the "driver" "passenger" nomenclature, that is commonly used in the cancer field. Conceptually, the "conditionally neutral" allosteric mutations might be viewed as "latent driver" mutations. "Latent drivers" may emerge prior to and during cancer evolution and do not manifest a noticeable gain of function (PMID: 25661093).

The role of other Ras related pathways, PI3K, RAL, RAC/RHOA, etc., which are all pathways to be shown to be extremely important (and in some cases directly affects the capability of the MAPK pathway to drive growth) should be discussed, since they may lead to "contradictory imbalances" in RAS function.

The same applies to other interactions of Ras with membranes and to the possible dimerization/clustering of Ras.

The interpretation of the results should take into account the differences in the relative binding affinity of GTP vs. GDP, protein stability or aggregation affected by mutations. It is also important to make clear the role of the activating mutants in terms of loss of GTPase activity (intrinsic or GAP-mediated) or increase in nucleotide dissociation (fast cycling mutants).

Along this line, regarding the residues that contact the nucleotide. it is well known that mutations of such residues lead to a loss of affinity. They do not typically alter the extent of loading considering the high affinity of wild-type protein and concentration of nucleotide. However, such mutations have a dramatic effect on the biochemistry and biology of Ras: they create fast-cycling mutants which in principle behave like GTPase-negative mutants in that they activate the protein. At the same time some of these become unstable, in particular in bacteria. A good example is Asp119 (Figure 2—figure supplement 1) where all the mutations lead to severe loss of growth which indicates that they are unstable.

The use of some previously reported X-ray structures that are questionable weakens the interpretation of the R to T state transition considerably.

The chapters on Rap proteins and Ras drug binding seems to be unnecessary and not to the point.

The emergence of a positive feedback loop in the activation of Ras by SOS, is interesting but lacks direct evidence to support the claim.

The presence, or not, of a SOS-like protein in *S. rosetta* has to be better clarified for the interpretation of the experiments.

The small reduction in hydrolysis rate for G12V versus WT is based on an assay not reliable for measuring low rates. This issue should be clarified or removed.

For the ITC experiments, binding between 25 Ras mutants and c-Raf RBD the authors selected a scatter plot to compare the ΔΔG (ITC) vs. ΔE (cell growth) values, and claim a linear correlation fits the data in the scatter plot. This is not obvious from the plot where many outliers are apparent. Furthermore, experimental details are missing, i.e. How was loading of Ras with GMPPNP verified? Given that the Ras/RBD affinity obtained from these experiments is much lower than reported by others, and the concentration of the Ras and RBD is also significantly lower than other publications, the results might be related with a higher salt concentration? The raw data from ITC experiments should be shown.

Raw data/corresponding errors analyses for some experiments are missing.

---

## [Author Response]

[…] The outcome is what we would intuitively expect: the allosteric switching function – particularly of a key hub in the cellular network like Ras – restricts the sequence. Most mutations mildly reduce the function of Ras. More severe effects are at residues that are functionally critical, such as binding to GTP, to Raf-RBD and to regulators.

We appreciate that one central finding of our paper, that there is a global constraint on the sequence in the context of the regulators, could have been anticipated from general considerations, but we note that a systematic analysis of this aspect of Ras has not been carried out previously. We feel that the other key finding of our paper is unexpected. This is the identification of strong gain-of-function (GOF) mutants that emerge in the context of removing the regulators – these hotspots of activation constitute a set of residues that is much broader than the much smaller set of known activating mutations in Ras. Thus, the review comment highlighted above pertains only to one of the experiments described in our paper (the regulated-Ras experiment).

The paper provides a large body of data on the activity of the variants obtained by mutation scanning. All this information is accompanied with a detailed structural interpretation, and complemented with a new X-ray structure and MD simulations.

This comprehensive analysis provides an anatomical sequence-structure-function map of a protein that controls critical cellular proliferation pathways. In summary, the data is very interesting and valuable.

Unfortunately, the interpretation of the data left many open questions. Indeed, the overall goal of the paper, beyond the detailed description of some of the mutants, is not completely clear.

We thank the reviewers for pointing out what we now recognize to be an obvious failing of our original manuscript. We recognize that the previous version of the manuscript had mixed up general concepts of sequence tolerance in proteins with issues that are specifically important for Ras, most importantly in the Abstract and in the Introduction. The concluding remarks section of the paper had failed to present a consolidated summary of the key results.

We hope that the revised manuscript now addresses these issues in a satisfactory manner. We have reorganized and rewritten the Abstract, Introduction and concluding section of the paper to make the overall goals of the paper clear, and to highlight the key discoveries. The principal goal of this paper is to delineate the constraints on the sequence of Ras imposed by the GTPase cycle and the action of the GAPs and the GEFs. We find that in the presence of the GAPs and the GEFs, the sequence of Ras is subject to a global constraint that suppresses mutations. This is very different from results obtained previously by saturation mutagenesis for proteins with simpler binding functions. A quite unexpected result of our work is the discovery of hotspots of mutation that activate Ras in the absence of regulators, and this is another important outcome of our work.

The relations between the proposed hot spots, the interactions with effectors and the molecular details of the allosteric mechanism, are not addressed clearly in the paper and the relation of the effectors with the intrinsic allosteric mechanism will have to be described explicitly.

Again, we recognize a failure on our part to provide proper context for the work. In the rewritten Introduction we provide a brief summary of the allosteric mechanism of Ras, pointing the reader to an insightful review by Vetter and Wittinghofer that should provide the interested reader with all the relevant details.

We note that we have provided numerical fitness data for every mutation as supplementary data files, and we hope that the interested reader will obtain these data. There is a rich literature on the allosteric mechanism of Ras, and by mapping the numerical data on to the available structures the interested reader should be able to obtain a complete understanding of our results.

The arguments around the possible evolutionary path to the acquisition of the GTPase regulation in vertebrates are not very strong, and the underlying analysis of the sequences does not seem to be at the same quality level than the rest of the paper. The direct relation between evolutionary analysis and the allosteric control in Ras should be clarified.

Regarding the quality of the ancestral sequence reconstruction, we have provided more information on how the sequences were generated, and also provided more information on the robustness of the reconstructions. This information is provided in the revised Materials and methods, in the relevant section. One point that we did not make clearly in the original manuscript is that our conclusions regarding the change in allosteric control are not dependent on the specific hypothetical ancestral sequence that is used. Indeed, similar conclusions can be drawn by using any invertebrate Ras sequence for comparison to the human H-Ras sequence. We now provide a new supplementary figure (Figure 8—figure supplement 1), in which the sequence of a choanoflagellate Ras is compared to that of human H-Ras, along with the fitness effects of the substitutions as seen in the unregulated Ras experiment. The conclusion is that replacement of residues in human H-Ras by the corresponding residues in the invertebrate Ras leads to activation of human H-Ras. It is, however, conceptually better to use the hypothetical ancestral Ras sequence for this comparison.

Regarding the possible evolutionary path to the modification of the GTPase regulation in Ras, we recognize that this part of the paper is speculative at present. The revised manuscript makes it clear that this section is speculative, by so stating at the beginning of the section headed “The variable regions of Ras are involved in interactions with the Ras activator Son-of-Sevenless”. We have shortened this section by removing the discussion of Rap. We retain the discussion of SOS, but have edited the text to point out more clearly that the significance of the “R” and “T” conformations is still not clearly understood, and have shortened this section as well.

Other more specific comments include:

Terminology, like 'latchkey' which is not well defined and 'allostery' quite loosely. The 'latch-like' interaction terminology is also confusing.

We agree. We have removed the use of “latch” or “latch-like” to describe residues, using mutational “hotspot” instead. In the revised manuscript we restrict the use of “allostery” or “allosteric” to a minimum (these words appear in only 12 places in the revised manuscript, where we feel that their use is justified by prior investigations into Ras).

The protein names, i.e. Ras, Raf and Rap are used as general terms, without reference to isoforms. Isoforms of Ras, Rap and Raf differ in sequence and have distinct signaling properties, so it is unclear why the investigators do not address isoform differences.

We thank the reviewers for pointing out this important issue. In the first paragraph of the paper we now use the term “Ras proteins” rather than “Ras”, and we then identify specific isoforms in the second paragraph. Throughout the manuscript, whenever it is important, we refer to human H-Ras specifically. We have now omitted discussion of Rap altogether.

Consider adopting the "driver" "passenger" nomenclature, that is commonly used in the cancer field. Conceptually, the "conditionally neutral" allosteric mutations might be viewed as "latent driver" mutations. "Latent drivers" may emerge prior to and during cancer evolution and do not manifest a noticeable gain of function (PMID: 25661093).

We appreciate the value of this comment, but we prefer at this time to avoid interpretation of our mutations in terms of cancer mechanisms. We have, however, added a sentence about “latent driver” mutations, and the stated reference, in the fifth paragraph of the subsection “Concluding Remarks”.

The role of other Ras related pathways, PI3K, RAL, RAC/RHOA, etc., which are all pathways to be shown to be extremely important (and in some cases directly affects the capability of the MAPK pathway to drive growth) should be discussed, since they may lead to "contradictory imbalances" in RAS function.

In the revised Introduction, we now state that Ras proteins can activate a multitude of pathways, in the first paragraph. We restrict discussion to Raf, however, because in the context of our experiments this was the only effector that was analyzed.

The same applies to other interactions of Ras with membranes and to the possible dimerization/clustering of Ras.

We have a statement concerning this in the fourth paragraph of the subsection “Concluding Remarks”.

The interpretation of the results should take into account the differences in the relative binding affinity of GTP vs. GDP, protein stability or aggregation affected by mutations. It is also important to make clear the role of the activating mutants in terms of loss of GTPase activity (intrinsic or GAP-mediated) or increase in nucleotide dissociation (fast cycling mutants).

This comment brings up a number of important points, not all of which can be addressed rigorously at this time. We have, however, included comments about these facts (subsection “Ras exhibits a global sensitivity to mutation when regulated by a GAP and a GEF”; subsection “Hotspots of activating mutations in unregulated wild-type Ras” and subsection “Hotspot residues dampen the conformational dynamics of wild-type Ras”.

Along this line, regarding the residues that contact the nucleotide. it is well known that mutations of such residues lead to a loss of affinity. They do not typically alter the extent of loading considering the high affinity of wild-type protein and concentration of nucleotide. However, such mutations have a dramatic effect on the biochemistry and biology of Ras: they create fast-cycling mutants which in principle behave like GTPase-negative mutants in that they activate the protein. At the same time some of these become unstable, in particular in bacteria. A good example is Asp119 (Figure 2—figure supplement 1) where all the mutations lead to severe loss of growth which indicates that they are unstable.

We have addressed this point in the subsection “Ras exhibits a global sensitivity to mutation when regulated by a GAP and a GEF” regarding the reduction in affinity of certain Ras variants at Asp 119, Lys 117, and Lys 147. While this point comes up during discussion of the regulated-Ras experiment, discussion of fast-cycling Ras variants is more pertinent to the section on hotspot mutations. We have edited the text accordingly to clarify this point. See subsection “Ras exhibits a global sensitivity to mutation when regulated by a GAP and a GEF”, subsection “The oncogenic G12V mutation attenuates the effects of mutations that activate Ras in the wild-type background”, and subsection “Hotspot residues dampen the conformational dynamics of wild-type Ras”.

The use of some previously reported X-ray structures that are questionable weakens the interpretation of the R to T state transition considerably.

We appreciate that the functional significance of the R and T states of Ras-GTP have not been established experimentally. We now say, in the third paragraph of the subsection “The variable regions of Ras are involved in interactions with the Ras activator Son-of-Sevenless”, that the functional significance of these states have not been established definitively. Nevertheless, we do find it interesting that one of the hotspot mutations (L120A) switches Ras from the “R” to the “T” state in our crystal structures, and also that the wild-type choanoflagellate Ras-GTP structure is in the T state rather than the R state seen for human Ras. We recognize that these are not definitive findings (crystal structures cannot be definitive about thermodynamic balances), but we hope that the interested reader will find these to be intriguing observations.

The chapters on Rap proteins and Ras drug binding seems to be unnecessary and not to the point.

We have removed discussion of Rap and Ras drug binding.

The emergence of a positive feedback loop in the activation of Ras by SOS, is interesting but lacks direct evidence to support the claim.

We agree that this section is speculative. We now explicitly state that the discussion is speculative, subsection “The variable regions of Ras are involved in interactions with the Ras activator Son-of-Sevenless”. Nevertheless, we do find it intriguing that the four regions of sequence divergence between the hypothetical ancestral Ras sequence and human H-Ras map correlate to features in the interaction of Ras with SOS. In recognition of the validity of this comment, we have shortened this discussion considerably. See the aforementioned subsection.

The presence, or not, of a SOS-like protein in S. rosetta has to be better clarified for the interpretation of the experiments.

We have removed discussion of *S. rosetta* SOS. The biochemistry of SOS is tangential to this manuscript, and a discussion of experiments on *S. rosetta* SOS would take us too far afield.

The small reduction in hydrolysis rate for G12V versus WT is based on an assay not reliable for measuring low rates. This issue should be clarified or removed.

We have clarified the limitations of the assay we used to measure intrinsic GTP hydrolysis rates of the G12V mutant. Slow rates of GTP hydrolysis are difficult to measure using a sensor of inorganic phosphate, and we have cited work from Ikura that accurately measures GTP hydrolysis rates by NMR in our discussion of gain of function mutations. See subsection “The oncogenic G12V mutation attenuates the effects of mutations that activate Ras in the wild- type background”, second paragraph.

For the ITC experiments, binding between 25 Ras mutants and c-Raf RBD the authors selected a scatter plot to compare the ΔΔG (ITC) vs. ΔE (cell growth) values, and claim a linear correlation fits the data in the scatter plot. This is not obvious from the plot where many outliers are apparent. Furthermore, experimental details are missing, i.e. How was loading of Ras with GMPPNP verified? Given that the Ras/RBD affinity obtained from these experiments is much lower than reported by others, and the concentration of the Ras and RBD is also significantly lower than other publications, the results might be related with a higher salt concentration? The raw data from ITC experiments should be shown.

We agree that the section on comparing the bacterial growth rate data to thermodynamic data was not written as carefully as it could have been. We have amended the text as shown below. Note that the linear correlation is not important – we do not derive thermodynamic inferences from the growth rate data. In the amended text below we clarify that the correlation is good for mutations with large effects. The effects of GTP hydrolysis and exchange, which do not enter into the ITC measurements, may account for the lack of correlation for small changes in binding affinity.

“We compared the relative fitness values, derived from two-hybrid experiments without the GAP or the GEF, to changes in the binding free energy for the interaction of Raf-RBD and Ras•GTP, measured using isothermal titration calorimetry (ITC). […] For mutation with near wild-type binding, there is a greater variance of relative fitness, a result that might arise due to effects on GTP hydrolysis and exchange rates, which do not enter into the thermodynamic measurement.”

We have added a sentence in the fourth paragraph of the subsection “Quantifying the effects of Ras mutations on function” stating that GMP-PNP loading of Ras variants was confirmed by HPLC. We have added a detailed protocol to the Materials and methods section that describes how the HPLC analysis was performed. A supplemental figure (Figure 1—figure supplement 4) is now provided that shows the complete loading of Ras with GMP-PNP.

We have provided references in the aforementioned paragraph from Wohlgemuth et al., Rudolph et al., and Kiel et al. in which Ras:GMP-PNP binding to the Raf-RBD was measured by ITC (see extracted

text above). These studies measured the affinity of the interaction as being approximately 100-120 nM, which is within the range of our measurement of the affinity at ~170 nM. Furthermore, the salt concentrations in our experiments (150 millimolar NaCl) are consistent with the cited literature, although the Ras:Raf-RBD interaction is indeed strongly dependent on salt concentration.

Raw data/corresponding errors analyses for some experiments are missing.

We assume that this comment refers to the raw data for the ITC experiments, and the measurements of GTPase rates and nucleotide exchange rates. These data are now provided for an extensive subset of the experiments.